# RA-induced prominence-specific response resulted in distinctive regulation of Wnt and osteogenesis

Chao Song[1], Ting Li[1], Chunlei Zhang[3], Shufang Li[1], Songhui Lu[2], Yi Zou[1]

Proper retinoic acid (RA) signaling is essential for normal craniofacial development. Both excessive RA and RA deficiency in early embryonic stage may lead to a variety of craniofacial malformations, for example, cleft palate, which have been investigated extensively. Dysregulated Wnt and Shh signaling were shown to underlie the pathogenesis of RA-induced craniofacial defects. In our present study, we showed a spatiotemporal-specific effect of RA signaling in regulating early development of facial prominences. Although inhibited Wnt activities was observed in E12.5/E13.5 mouse palatal shelves, early exposure of excessive RA induced Wnt signaling and Wnt-related gene expression in E11.5/E12.5 mouse embryonic frontonasal/maxillary processes. A conserved regulatory network of *miR-484-Fzd5* was identified to play critical roles in RA-regulated craniofacial development using RNA-seq. In addition, subsequent osteogenic/chondrogenic differentiation were differentially regulated in discrete mouse embryonic facial prominences in response to early RA induction, demonstrated using both in vitro and in vivo analyses.

## Introduction

Retinoic acid, the metabolite of vitamin A, is the natural ligand of nuclear RA (retinoic acid) receptors and regulates the transcription of genes containing the RA response element (RARE). By recruiting either nuclear receptor coactivators or nuclear receptor corepressors, RA-RARs can directly activate/repress the transcription of key developmental genes such as *Hox* genes in hindbrain formation, *Sox* genes in neuromesodermal progenitor differentiation, and *Fgf8* in developing heart (Ghyselinck & Duester, 2019). Despite the discrepancies, both gain-of-function studies and genetic loss-of-function studies indicate that the dysregulated RA signaling has teratogenic effects and result in aberrant development of hindbrain, body axis formation, heart patterning, and eye morphogenesis (Berenguer et al, 2022).

Expression of retinaldehyde dehydrogenases, which is responsible for RA synthesis, starts at E7.5 mouse embryo in presomite mesoderm. The first embryonic RA signaling was detected and restricted within the developing eye around E8.5 (Dubey et al, 2018). Excessive RA signaling during early embryonic development were shown to affect the fate of cranial neural crest cells (CNCCs) and resulted in craniofacial malformations including the cleft lip/palate (Williams & Bohnsack, 2019). Formation of the facial prominences appears around E8.5, upon receiving a great contribution of neural crest cells (NCCs) delaminated from between the dorsal neural tube and the overlying ectoderm (Roth et al, 2021). The emigration and proliferation of NCCs, along with the paraxial mesoderm–originated mesenchymal cells, ensures the rapid growth of these facial prominences (Murillo-Rincón & Kaucka, 2020). By E11.5, the bilateral medial nasal prominences, lateral nasal prominences, and maxillary prominences are in close contacts and complete the fusion processes on each side of the developing face (Ji et al, 2020). Ongoing convergence of these opposite structures in the midline form a continuous sheath of the front face and the primary palate (Suzuki et al, 2018). Meantime, the mediolateral outgrowths of the maxillary form palatine processes starting around E11.5, growing vertically along the developing tongue in the mouth cavity (Suzuki et al, 2018). The growth and elevation of the palatine processes lead to their horizontal fusion on top of the tongue in the midline and fusion to the primary palate just behind the incisive foramen by the time of E15.5 (Martinelli et al, 2020). The mesenchyme derived from NCCs form ganglia, skeleton, and connective tissues, with musculature derived from paraxial mesoderm (Roth et al, 2021).

Underlying cellular events including cell proliferation, migration, apoptosis, and epithelial–mesenchymal transition of epithelium after fusion are well-synchronized processes, coordinated by key-signaling pathways, governing early morphogenesis. For instance, Wnt acts upstream of *Bmp4* and induces its expression in overlapping domains (Reynolds et al, 2020). The activation of Wnt and Bmp signaling is observed in the pre-fusion epithelia and the underlying mesenchyme of facial prominence to ensure proper mesenchymal growth, apoptosis, and epithelial fusion (Jiang et al, 2006). However, excessive activation of Bmp signaling also causes craniofacial malformation. Knockout of noggin (*Nog*), an antagonist

[1]The Key Laboratory of Virology of Guangzhou, Jinan University, Guangzhou, China [2]Southern Marine Science and Engineering Guangdong Laboratory (Zhuhai), Zhuhai, China [3]First Affiliated Hospital, Jinan University, Guangzhou, China

Correspondence: tyizou@jnu.edu.cn

 

of Bmp pathway, displayed increased Bmp/Smad activity in E13.5 palate epithelium and resulted in a complete cleft palate in mice (Reynolds et al, 2020). Although excessive intake of vitamin A and exogenous RA were shown to induce craniofacial malformation in humans and mice, the molecular pathogenesis associated with altered RA signaling was poorly understood (Lammer et al, 1985; Rothman et al, 1995; Ackermans et al, 2011). Inhibitory effects of RA on Tgf-$\beta$ and Wnt signaling, whereas role of RA in maintaining *Shh* and *Fgf8*, were demonstrated in various cultured cell lines and in RA-induced CL/P in mouse and chick models (Roa et al, 2019; Reynolds et al, 2020; Wu et al, 2022). However, controversial results showing RA-induced down-regulation of *Shh* expression were also presented (El Shahawy et al, 2019; Wang et al, 2019). The differences in animal models used and in the strategies of RA application may partially explain the discrepancies.

In this work, we investigated the excessive RA-induced changes in gene expression profile in the developing craniofacial prominences using mouse model. Multiple dosages of all-trans RA (*at*RA) were applied to pregnant mice between early gestation (from E8.5 to E10.5), which was reported to induce full penetrance of CP (Wang et al, 2019). The frontonasal prominence (FnP), a pair of maxillary processes (MxP) and a pair of mandibular processes (MdP) at E12.5 were collected and subjected to RNA-sequencing. The enrichment analysis of transcriptomic changes indicated the altered gene expression of 12 Wnt signaling components. A significantly down-regulated miRNA, *miR-484*, targeting five out of these 12 Wnt molecules, was predicted as a hub driver. Further screening for differentially expressed genes (DEGs) related to craniofacial malformations that were transcriptionally regulated by Wnt signaling revealed *Foxn3* and *Itgb1*, a member of fork head family of transcription factors that regulate osteogenic gene expression and integrin $\beta$1 that regulate osteoblast differentiation. The gene expression of our bioinformatic analyses and transcriptome results were experimentally validated in prominence-specific fashion using quantitative qRT–PCR. The RA-induced different patterns of change in Wnt activities were assessed in craniofacial mesenchymal cells derived from various facial prominences at different stages using TCF/LEF-reporter TOP/FOP-flash assay. The subsequent influence of RA exposure on osteogenic differentiation was investigated both in vitro using mouse embryonic craniofacial mesenchymal cells derived from different facial prominences and in vivo using RA-induced mouse embryos. The results suggested a spatiotemporal-specific effect of early RA signaling in regulating Wnt activity and osteogenic/chondrogenic differentiation in discrete embryonic facial prominences.

# Results

## Early prenatal RA exposure induce developmental defects including midfacial anomalies

Prenatal RA exposure affects embryonic development in a both dose-dependent and embryonic stage–dependent manner. Oral administration of 25 µg/g body weight *at*RA consecutively at E8.5, E9.5, and E10.5 gestational timepoints was performed according to

the description from Wang et al (2019). Full penetrance of CP was observed, in line with the observation reported by Wang et al (2019). Consistent with previous studies (Berenguer et al, 2018), gross morphological analysis between E12.5–E15.5 revealed a range of malformations, resulting from defective early RA signaling, including microphthalmos, limb reduction, and superficial hemorrhages, with various incidences (Fig 1A–G').

For transcriptomic profiling, the FnP, bilateral MxP and MdP were micro-dissected from fetuses at E12.5, at which stage the craniofacial abnormalities induced by RA could be easily verified (Fig 1C). E12.5 is also the turning point from the completion of facial development to the beginning of palate formation, with the vertical palatal shelves (PS) being easily distinguished on both sides of the MxP. To minimize the chance of technical variance in sampling, each fetus was collected from an independent conception with (n = 4) or without RA exposure (n = 4). The dysregulated craniofacial development was confirmed by morphological analysis before sampling.

### An overview of the transcriptome and quality assessment

94.9% of the mRNA-seq datasets and 98.7% miRNA-seq datasets, respectively, remained after the initial filtering. The medians of log-expression (log-count-per-million) of each sample were presented post filtering using threshold of count-per-million >0.8 (Fig 2A and B, Table S1). Genes were included in the analysis if their expression were identified in at least four samples with reads count >0, detected by filterByExpr function of edgeR (version 3.14.0). In the mRNA expression datasets, >97% of the paired-end raw reads were mapped using the mm (10) reference genome (GRCm38.p4). Of the 34,754 total genes, 17,024 (49.0%) gene expressions were detected in all samples. In the miRNA expression datasets, >80% of the paired-end raw reads were mapped using the reference miRBase (version 20.0). Of the 1,907 total genes, 835 (43.8%) gene expressions were detected in all samples. The expression of *Zic3* was below the threshold in the mRNA expression datasets of all samples, excluding the potential contamination from adjacent forebrain tissue (Inoue et al, 2007).

The RA induced divergence in overall gene expression was demonstrated by the clustering of gene expression data among treatments in principal component analysis (Fig 2C and D). We identified a total of 1,047 DEGs (525 up-regulated and 522 down-regulated) and a total of 76 differentially expressed miRNAs (DEMs) (54 up-regulated and 22 down-regulated) with statistical significance (one-way ANOVA, adjusted *P*-value <0.05; Fig 2E and F, Tables S2, and S3). The agglomerative hierarchical clustering of the DEG and DEM using Pearson correlation coefficient and average distance also supported the dominant differences between treatments. Cluster analysis with approximately unbiased (AU) *P*-values and bootstrap probability (BP) values showed that two main groups were identified, significant at least at the 95% CI (*P* <0.05, Fig 2G and H).

The RA-synthesizing enzymes, Rdh10, Aldh1a2, and Aldh1a3 were abundantly expressed, and Cyp26b1 turned out to be the dominant subtype of enzyme for RA degradation, in agreement with the expression pattern of these genes in previous transcriptomic analysis of mouse embryonic craniofacial tissues (GSE62214 and

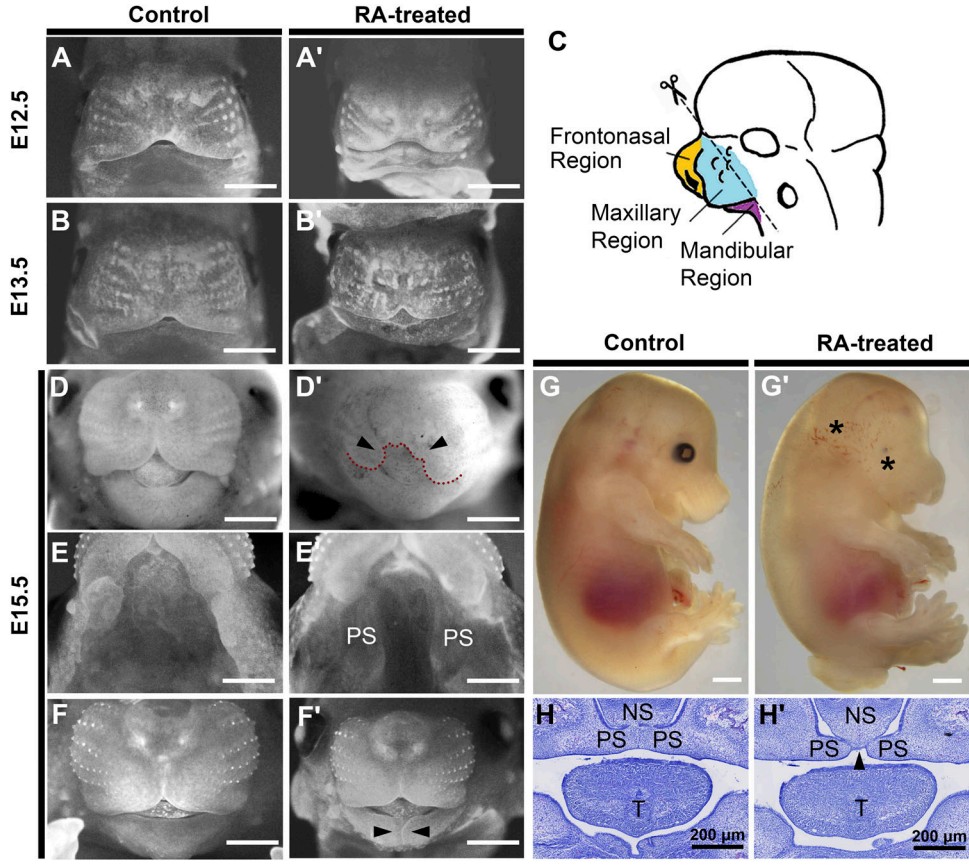

**Figure 1. Abnormalities at different developmental stages induced by early RA exposure.**
Representative images of some of the malformations resulted from early RA exposure. **(A, A', B, B', D, D', E, E', F, F', G, G', H, H')** Whole-mount nuclear fluorescent imaging of control (A, B, D, E, F, G, H) and RA-treated (A', B', D', E', F', G', H') embryos between E12.5 and E15.5, displaying hypoplasia of frontonasal prominence (FnP) (A'), reduced growth of medial nasal prominence (B'), cleft of upper lip (D'), cleft palate (E'), and cleft in mandible (F'). Scale bars: 1 mm; PS, palatal shelf. **(G, G')** Gross morphology of E15.5 control (G) and RA-treated (G') mouse embryos. * indicate microphthalmos and superficial hemorrhages. Scale bars: 1 mm. **(H, H')** H&E staining the coronal sections through the palatal regions of E15.5 embryos, showing failed fusion of PS in RA-treated mice, indicated with an arrowhead. Scale bars: 200 μm; NS, nasal septum; PS, palatal shelf; T, tongue. **(C)** Schematic of E12.5 mouse embryonic head. Dotted line illustrated the FnP, bilateral maxillary processes and mandibular processes that were dissected for transcriptomic profiling. FnP, frontonasal prominence; MxP, maxillary processes; MdP, mandibular processes.

GSE7759) (Fig S1). In addition, our results, along with the others, showed that mouse embryonic face expressed all isoforms of RA receptors including nuclear RA receptors (Rarα, Rarβ, Rarγ) and retinoid X receptors (Rxrα, Rxrβ, Rxrγ) (Fig S1). *Rxrγ* was expressed at a lower level in the epithelium and mesenchyme of all facial prominences between E10.5 and E12.5, although relatively abundant *Rxrγ* was found in MdP (GSE7759), primarily in the mandibular mesenchyme (GSE62214) (Feng et al, 2009; Hooper et al, 2017). No significant differences in the expression of these genes were noticed between the control mouse embryos and RA-induced mouse embryos (Fig S1A).

## Functional annotations and enrichment analysis revealed anomalous Wnt-related gene expression in RA-treated embryonic craniofacial prominences

Given the heterogeneity of the samples with respect to the tissue compositions and sex differences (with three females and one male in the control group whereas one female and three males in the RA-induced group), small fold changes might still be biologically important. Therefore, all DEGs with statistical significance were functionally annotated and undertaken enrichment analysis with no fold change cutoffs. Considering our particular interest in the early embryonic signaling, the Gene Ontology biological process terms (GO-BP terms) and Kyoto Encyclopedia of Genes and Genomes (KEGG) pathway enriched in genes differentially expressed

were analyzed in up-regulated and down-regulated genes induced by RA exposure, respectively. The enriched GO-BP terms were identified with DAVID functional annotation (DAVID v2022q2) which include DAVID Gene Functional Classification Tool and DAVID Functional Annotation Clustering Tool. The resulting enriched terms were ranked by gene count and the Wnt-signaling–associated terms were the only specific developmental pathway recognized in up-regulated DEGs with adjusted *P* <0.05 (Fig 3A, Table S4). No over-represented categories of GO-BP terms was identified in the down-regulated DEGs (Fig S2A). Furthermore, KEGG pathway analysis also recognized the over-represented genes involving Wnt signaling in the up-regulated DEGs, with 2.79-fold enrichment at adjusted *P* <0.05 (Fig 3B, Table S5). Down-regulated DEGs lacked enrichment in any categories of specific developmental signaling pathways (Fig S2B).

## Identification of mouse-specific miRNA–gene interactions targeting the up-regulated genes in Wnt pathway in DEMs

First, we searched TargetScan 8.0 and miRWalk for potential miRNAs–gene interactions for the 12 up-regulated genes enriched in Wnt signaling in KEGG pathway analysis. miRNAs were only considered as potential regulators of these genes when predicted in both databases, and as a result, 1,584 miRNAs and 5,473 miRNA–gene interactions were initially identified (Table S6). Next, the overlapping miRNAs between these potential miRNA regulators and the RA-induced down-regulated DEMs were selected (Fig S3A). 26

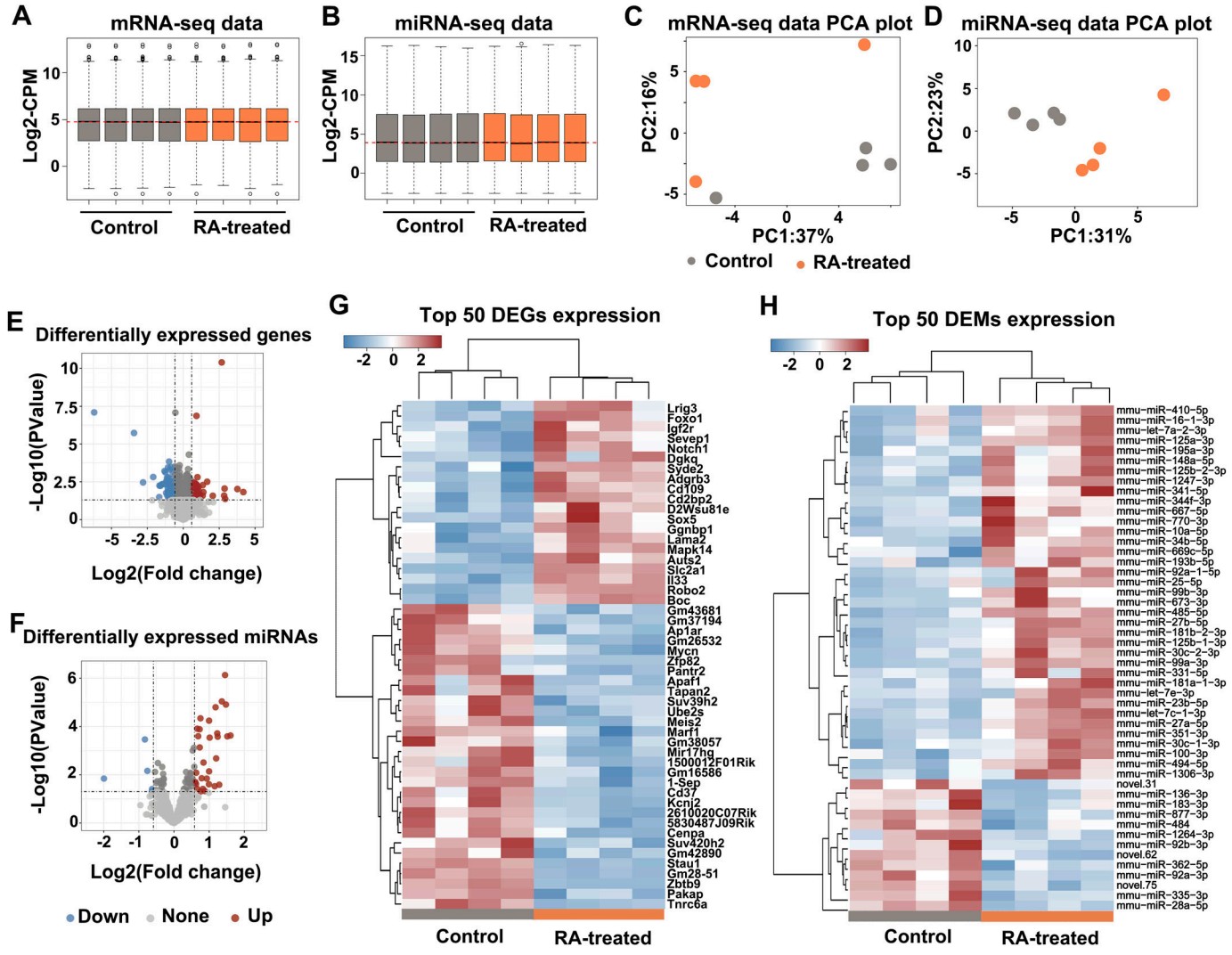

**Figure 2. Early RA exposure induced altered gene expression in developing mouse face.**
**(A, B)** Boxplots of log-count-per-million values showing expression distributions for normalized data of mRNA-seq and miRNA-seq. **(C, D)** Principal component analysis of the facial gene expression of control (grey) and RA-treated (orange) E12.5 mouse embryos. **(E, F)** Volcano plot of the differentially expressed genes and differentially expressed miRNAs between control and RA-treated mouse embryos. The horizontal dashed line represents a *P*-value of 0.05, and the vertical dashed lines represent fold changes of –1.5 and 1.5, respectively. **(G, H)** Heatmap of hierarchical clustering dendrograms of top 50 differentially expressed genes and differentially expressed miRNAs in control and RA-treated mouse embryos. Control samples. Both mRNA and miRNA of two groups were clustered with approximately unbiased (AU) *P*-values and bootstrap probabilities more than 95%. Color corresponds to the relative expression levels, red for up-regulated and blue for down-regulated genes, respectively.

nodes (14 DEMs and 12 Wnt genes) and 38 edges with 47 MREs in the mRNA of these 12 Wnt-related genes were identified (Fig 4A, Table S7). According to the maximal clique centrality (MCC) score calculated using cytoHubba plugin (version 0.1), miR-484 (MCC = 5) were identified as a hub miRNA. Four miR-484 binding sites on *Fzd5* 3′UTR were predicted by TargetScan (Fig S3B) and the miR-484: *Fzd5* mRNA association was further characterized using luciferase assay. 515 bp murine *Fzd5* 3′UTR containing either three clustered WT seed sequences of miR-484 or sense mutations of all these three seed sequence was cloned into *Sac* I/*Xba* I site of pmirGLO vector, respectively (Fig S3C). Reduced luciferase activities were observed in primary mouse embryonic mesenchymal cells and in HEK-293T cells co-transfected with pmirGLO-WT-*Fzd5* 3′UTR and miR-484 mimics, whereas no significant change was observed in cells co-transfected

with pmirGLO-MUT-*Fzd5* 3′ UTR and miR-484 (Figs 4B and S3D). Scramble miRNA transfections were used as controls.

The RA-induced change of expression of miR-484 was subsequently verified in vivo using qRT-PCR. Interestingly, distinctive regulation of miR-484 expression was observed in different facial prominences, with RA-induced down-regulation in E12.5 FnP/MxP/MdP and RA-induced up-regulation in E12.5 palate shelves (Fig 4C). We further verified the prominence-specific regulation of miR-484 in vitro using cultured primary mesenchymal cells collected from each individual prominence. Reduced miR-484 expression in mesenchymal cells derived from E12.5 FnP/MxP/MdP whereas elevated expression of miR-484 in mesenchymal cells derived from E12.5 palate shelved were observed upon 1 *μ*M *at*RA induction (Fig 4D). 1,442 bp sequence of the proximal-promoter region from –1,437 to +5

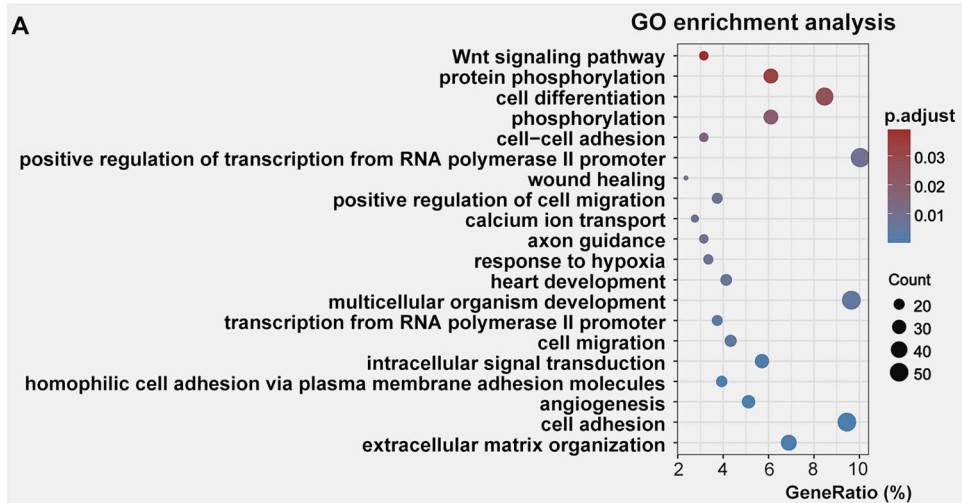

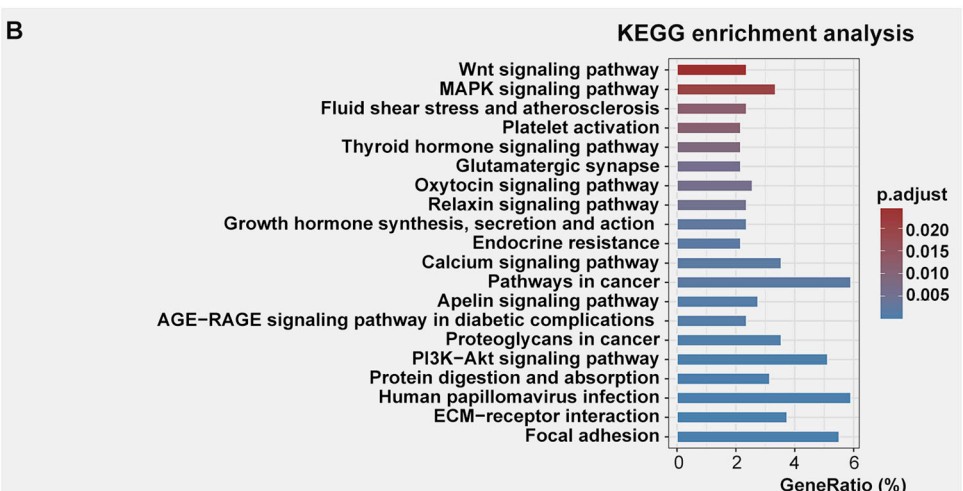

Figure 3. The top 20 Gene Ontology biological process terms and top 20 Kyoto Encyclopedia of Genes and Genomes pathways in enrichment analysis of up-regulated differentially expressed genes. (A, B) The colors of dots (A) and bars (B) indicate the adjusted *P*-value of the Fisher's exact test. The dot size indicates the number of genes, and the X-axis represents the percentage of differentially expressed genes identified in each term.

upstream with respect to the transcription start site of miR-484 gene was used to identify the potential presence of RARE. Two putative RAR-binding motifs were found at −40 to −29 and −190 to −179 (Fig 4E). A 500 bp promoter–enhancer region containing the WT or mutant sequence of RAREs was cloned into the vector pEZX-FR01, which were transiently expressed in HEK-293T cells and then induced with or without 1 μM *at*RA, respectively. The RA-induced fold change of the relative expression of luciferase driven by promoter containing mutated proximal RA cis-regulatory element (mut#1) was significantly reduced, compared with that driven by the WT promoter (Fig 4F). Mutation of both RAREs (mut#2) in the promoter resulted in further decreases in the RA-induced luciferase reporter gene expression. The results confirmed the predicted cis-acting motif in the proximal-promoter region of *miR-484* gene per se was RA responsive.

### RA enhanced Wnt signaling and induced Wnt-related gene expression during early craniofacial development

The Wnt activities were determined in E12.5 mouse primary craniofacial mesenchymal cells and in E12.5/E13.5 mouse embryonic palatal mesenchymal (MEPM) cells using Wnt reporter TOP/FOP-

flash. The mesenchymal cells were isolated from E12.5 MxP, E12.5 MdP, E12.5, and E13.5 PS, respectively, from the embryos of control and RA-treated pregnant mice. The MEPM cells collected from E13.5 mouse PS were used as references for palatal development that were widely used by others.

The elevated relative luciferase activities were observed in RA-treated E12.5 MxP mesenchymal cells compared with their untreated controls, whereas no significant change of Wnt activities was observed in E12.5 MdP mesenchymal cells (Fig 5A). On the contrary, reduced luciferase activities were observed in E12.5 and E13.5 MEPM cells collected from RA-induced mouse embryos. The results corroborated the RA-induced inhibition of Wnt signaling in the developing palate, as also indicated in previous studies by demonstrating RA-induced down-regulation of Wnt effector, *β*-catenin, in embryonic palate (Hu et al, 2013). Our results suggested that the embryonic facial prominence responded to RA-induction in a temporospatial-specific manner and indicated a distinctive patterning between early craniofacial and late palatal development. The expression of selected differentially expressed Wnt-related genes were validated using quantitative qRT–PCR in the correspondent E12.5 embryonic craniofacial tissues E12.5 (Fig S4). The expression of *Fzd5*, *Fzd9*, and

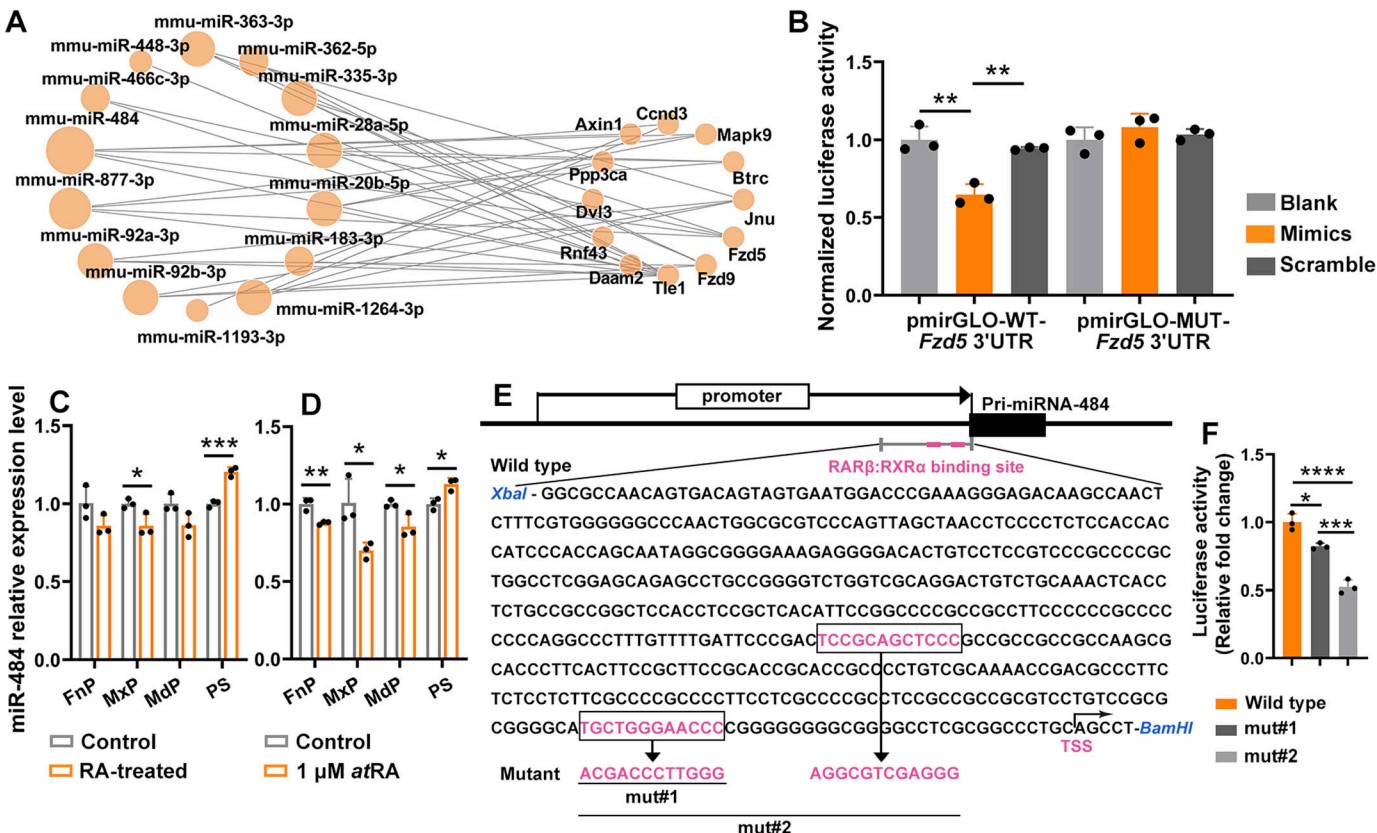

**Figure 4. miR-484 was recognized as the hub regulator.**
**(A)** miRNA–mRNA regulatory network of the down-regulated differentially expressed miRNAs and the up-regulated Wnt-related genes induced by RA. The size of miRNA nodes indicated the number of their target genes. **(B)** *Fzd5* was verified to be the direct target miR-484 using luciferase assay. Luciferase reporter construct contained three clustered WT seed sequences of miR-484 or sense mutations of all three seed sequences were co-transfected with miR-484 mimics or scramble RNA into primary mouse mesenchymal cells. Luciferase activities were measured 48 h post transfection. The relative luciferase activities were normalized to blank controls of mock transfection. The data were presented as means ± SD (n = 3). **P < 0.01, one-way ANOVA, Turkey's test. **(C)** The relative expression of miR-484 in each facial prominences of E12.5 embryos of RA-induced mice and controls. **(D)** The relative expression of miR-484 in cultured primary embryonic mesenchymal cells derived from E12.5 frontonasal prominence/maxillary processes/mandibular processes/PS, induced with or without 1 μM *at*RA. **(E)** The sequence of the miR-484 promoter containing WT or mutant retinoic acid response element, boxed and highlighted. The transcriptional start site was indicated. **(F)** Fold change of the expression of luciferase driven by the WT and the mutated promoter of miR-484 in response to 1 μM *at*RA induction. Values (C, D, F) were presented in means ± SD (n = 3). *P < 0.05, **P < 0.01, ***P < 0.001, two-tailed, unpaired *t* test.

their downstream effector, *Dvl3*, were further determined in each facial prominence over time course from E10.5 to E12.5 at 24 h intervals and in E13.5 palate shelves, respectively. The expression of *Fzd5* and *Dvl3* was up-regulated in RA-induced E12.5 FnP/MxP and down-regulated in E12.5 PS, whereas no significant changes in the expression of *Fzd5* and *Dvl3* in E12.5 MdP were observed (Fig 5B and C). However, early RA exposure did not induce any significant changes in *Fzd9* expression in all facial prominences at all developmental stages, except significant up-regulation in E10.5 MdP (Fig 5D). A trend of increasing in *Fzd5* and *Fzd9* expression over development, from E10.5 to E12.5, was noticed in all three facial prominences (Fig 5B and D).

## Identification of Wnt-regulated DEGs that associated with craniofacial malformation

Activation of canonical Wnt pathway results in the accumulation of cytoplasmic β-catenin and the acceleration of its translocation in nucleus, where it binds TCF/LEF family of transcription factors to

enhance target gene expression (Doumpas et al, 2019). Searching hTFtarget for the potential target genes regulated by TCF7 and LEF1, we identified 18 target genes that were also up-regulated in our transcriptome datasets (Table S8). Subsequently, the following key words: cleft lip or cleft palate or orofacial cleft or craniofacial phenotype, were used to search Mouse Genome Informatics database. The key words: cleft lip or cleft palate or orofacial cleft or craniofacial malformations were used to search MalaCards, a comprehensive automatically mined database of human diseases. We then obtained 1,751 genes related to facial malformations in either humans or mice after filtering redundant terms (Table S9). Among the 18 Wnt-regulated DEGs, we eventually identified two of them, *Itgb1* and *Foxn3*, associated with deregulated facial morphogenesis in human and mouse (Fig 6A). Foxn3 belong to the fork head family of transcription factors and has been implicated in regulating osteogenesis in craniofacial development (Samaan et al, 2010; Schmidt et al, 2011). Overlapping craniofacial defects were observed in mice with loss-of-function mutation of Foxn3 and in human patients with 14q32.11 deletion encompassing *FOXN3*. Itgb1

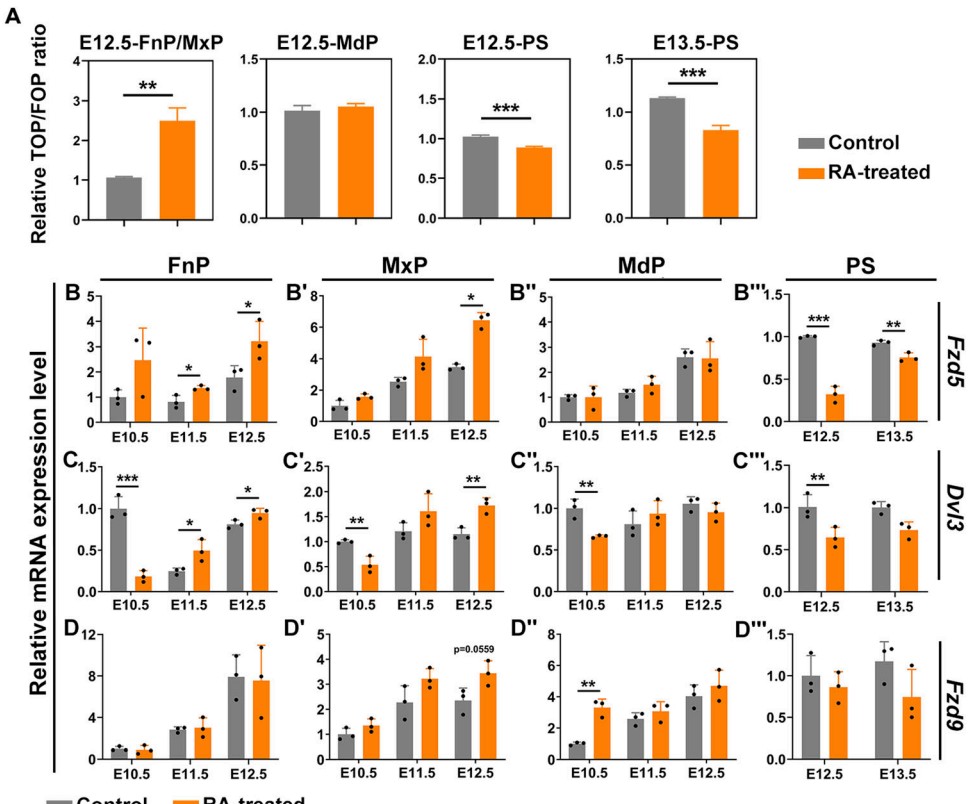

**Figure 5. RA induced Wnt activities and Wnt-related gene expression in frontonasal prominence/maxillary processes.**
**(A)** TOP/FOP luciferase reporter assays in the mesenchymal cells derived from each E12.5 craniofacial prominences and from E12.5/E13.5 PS of control and RA-treated embryos. Relative Wnt/β-catenin signaling activity was determined as a ratio of firefly/Renilla luciferase intensity. The data were presented as means ± SD (n = 3). **P < 0.01, ***P < 0.001, two-tailed, unpaired t test. **(B, C, D)** qRT-PCR of the expression of Wnt-related genes in each facial prominence and palate shelves of embryos between E10.5 and E13.5 of control and RA-treated mice. **(B, B″′, C, C″′, D, D″′)** The expressions of Fzd5 (B, B″′) Dvl3 (C, C″′) and Fzd9 (D, D″′) were referred as relative values of the expression of GAPDH and normalized to that of each E10.5 facial prominence from control mice. Gene expression in PS was normalized to that of the E12.5 PS from control mice. Values were presented in means ± SD (n = 3). *P < 0.05, **P < 0.01, ***P < 0.001, two-tailed, unpaired t test.

(integrin β1) is a member of the integrin superfamily and positively regulates Wnt signaling by inducing β-catenin, which in turn, activates Itgb1 expression (Gu et al, 2022). The expression of Itgb1 was shown to correlate with osteogenic differentiation via modulating Wnt signaling (Yang et al, 2020). The influence of RA exposure on the expression of Foxn3 and Itgb1 in each facial prominence was verified using quantitative qRT–PCR. Our results demonstrated the significantly elevated expression of Foxn3 in FnP and MxP of the E11.5 and E12.5 mouse embryos of RA-treated mice (Fig 6B and C). Non-significant trend of RA-induced increasing in Itgb1 expression was also observed in the FnP/MxP from E10.5–E12.5 (Fig 6E and F). The expression of Foxn3 and Itgb1 in MdP at all developmental stages were not affected by RA exposure compared with those of the controls (Fig 6D and G). We also noticed the trend of increasing expression of Foxn3 and Itgb1 in all prominences with respect to the developmental age, consistent with the tempospatial-specific increasing of Fzd5 expression and the osteoblasts/chondroblasts differentiation underlining the facial mesenchymal programing (Fig 6B–G). The expression of Foxn3 and Itgb1 was down-regulated/unchanged in E12.5 and E13.5 PS in RA-treated embryos compared with that of the controls, consistent with the pattern of the changes of Wnt activities in these tissues in RA-treated embryos (Fig 6H and I). The correlation analysis using our mRNA transcripomic datasets again revealed the positive correlation between the expression of Foxn3/Itgb1 and Fzd5, respectively, supporting a Wnt-mediated expression of Foxn3 and Itgb1 in early craniofacial development and a pivotal role of Wnt signaling in regulating mesenchymal differentiation (Fig 6J).

## Increased expression of osteogenic genes in craniofacial tissues and the enhanced osteogenesis in RA-treated mouse embryos

Both Itgb1 and Foxn3 null mutant mice displayed a range of abnormal craniofacial morphology, with cleft palate and abnormal ear development observed in Itgb1-KO mice, whereas defective development of the overall skull including frontal, parietal, mandible, palatal, and dental bones was observed in Foxn3-KO mice (Aszodi et al, 2003; Samaan et al, 2010). In addition, a number of osteogenic genes, including Bmp2, Bmp4, Bmp7, and RUNX2, were shown to be transcriptionally regulated by Foxn3 (Samaan et al, 2010). The expression of Bmp7 was also found to be increased in our transcriptomic profile of RA-induced embryos compared with that of the controls. We therefore verified the expression of Bmp7 in facial prominences using qRT–PCR. Enhanced expression of Bmp7 was observed in E10.5–E12.5 FnP/MxP, whereas reduced expression of Bmp7 was observed in E12.5–E13.5 PS, consistent with the inhibition of Wnt signaling and decreased Wnt-related gene expression in RA-induced embryonic palate tissues (Fig 7A). In addition, expression profiling of osteogenic and chondrogenic marker genes in the E12.5 embryonic facial tissues also revealed the prominence-specific pattern of differentiation in response to excessive RA signaling. The expression of osteoblast markers (Alp, Runx2, and Col1a1) was increased whereas the expression of cartilaginous markers (Sox9, Sox5, Sox6, Col2, and Col10a1) was decreased in the MxP of RA-treated embryos, compared with that of the control mouse embryos (Fig 7). On the contrary, reversed pattern of these gene expressions

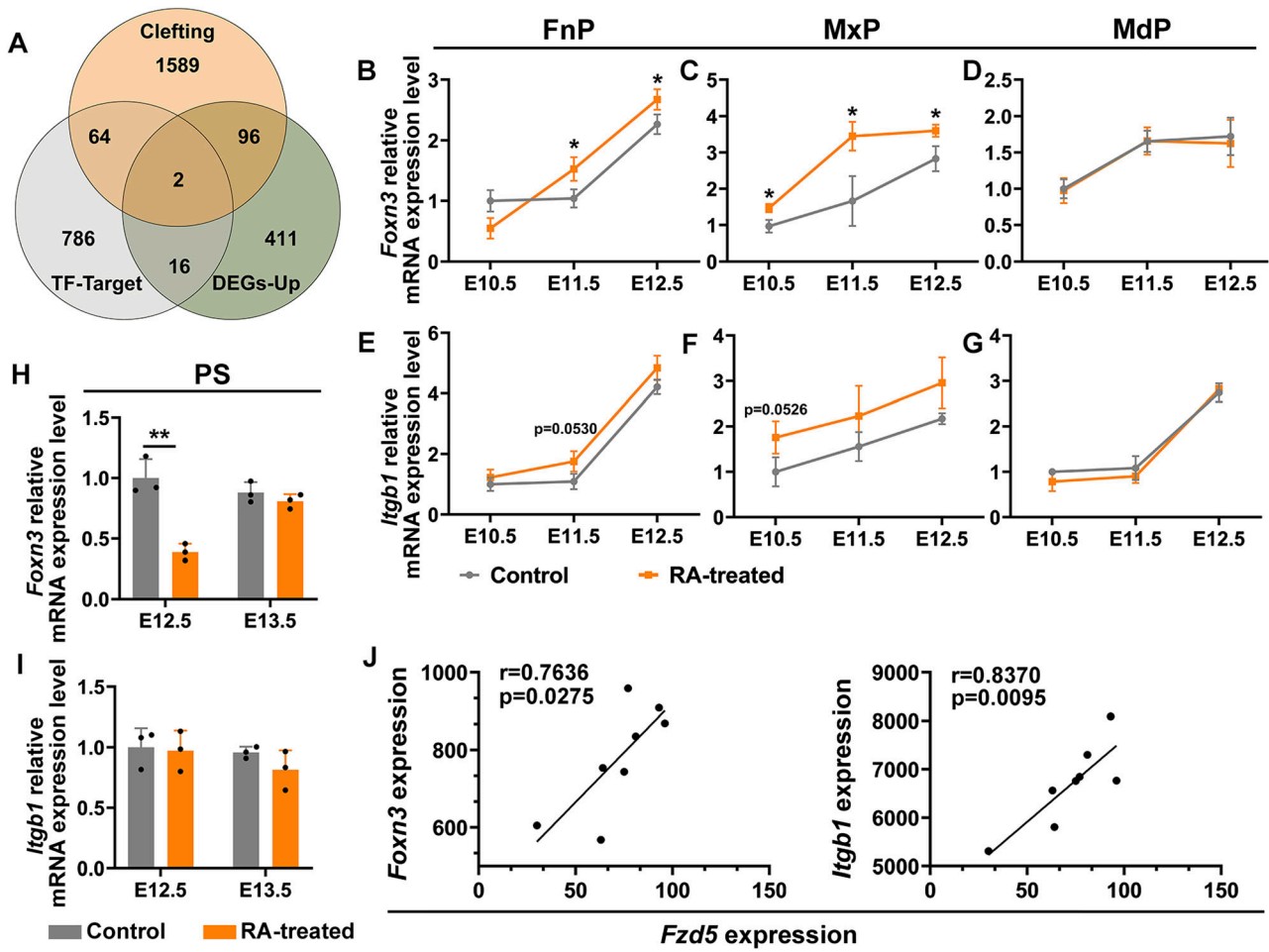

**Figure 6. Two Wnt-activated genes, *Foxn3* and *Itgb1*, cleft-associated, were induced in embryonic face by early RA exposure.**
**(A)** Venn diagram identified *Foxn3* and *Itgb1* in up-regulated differentially expressed genes were regulated by Wnt, recognized in hTFtarget databases, and were associated with deregulated facial morphogenesis, recognized in Mouse Genome Informatics and MalaCards databases. **(B, C, D)** qRT-PCR analysis showed the relative expression of *Foxn3* in frontonasal prominence (B), maxillary processes (C), and mandibular processes (D) from E10.5–E12.5, respectively. *P < 0.05, two-tailed, unpaired *t* test. **(E, F, G)** qRT-PCR analysis showed the relative expression of *Itgb1* in frontonasal prominence (E), maxillary processes (F) and mandibular processes (G) from E10.5–E12.5 mouse embryos of control and RA-treated mice, respectively. **(H, I)** qRT-PCR analysis showed the relative expression of *Foxn3* (H) and *Itgb1* (I) in E12.5 and E13.5 PS from control and RA-treated mouse embryos. **P < 0.01, two-tailed, unpaired *t* test. **(J)** Scattered plots of correlation between the expression of *Fzd5* and *Foxn3* or *Itgb1* in our mRNA-seq datasets. Pearson correlation analysis showed significant positive correlation (r = 0.7636, *P* = 0.0275) between the expression of *Fzd5* and *Foxn3*, and significant positive correlation (r = 0.8370, *P* = 0.0095) between the expression of *Fzd5* and *Itgb1*, respectively.

was observed in the palate shelves of RA-treated embryos, compared with that of the control mice (Fig 7B). We further collected the E12.5 maxillary mesenchymal cells and E12.5 MEPM cells from the control mouse embryos and cultured in osteogenic medium with or without 0.5 *μ*M *at*RA for analysis of osteogenesis in vitro. The expression of osteogenic genes was characterized using qRT–PCR. Our results showed that in vitro treatment of 0.5 *μ*M *at*RA also increased the early expression of *Alpl* and the late expression of *Col1a1* in cultured MxP mesenchymal cells, control cells treated with vehicles (Fig 7C). The expressions of *Alpl*, *Runx2*, and *Col1a1* were down-regulated in PS mesenchymal cells with 0.5 *μ*M *at*RA treatment, compared with that of the controls (Fig 7C). Alkaline phosphatase staining and alizarin red staining further confirmed RA-promoted osteogenic differentiation in maxillary-derived mesenchymal cells and inhibited osteogenic differentiation in PS-derived mesenchymal cells in vitro (Fig 7D–F). In addition, analysis of the phenotypes of

both chondrocranium and dermatocranium of E15.5 mouse embryos revealed the presence of palatoquadrates that was supposed to be found in primitive jawed vertebrates, in agreement of previous report (Vieux-Rochas et al, 2007) and enhanced osteogenesis in the proximal end of MxP/MdP in RA-induced mouse embryos (Fig 8A–F and A'–F'). These results suggested that the distinctive pattern of gene expression and differentiation in respond to RA induction was facial prominence-specific (Fig 8G).

## Discussion

Cleft lips with or without cleft palate are one of the most common congenital malformations in humans. However, cleft lip and cleft palate have distinctive pathological etiology due to the different

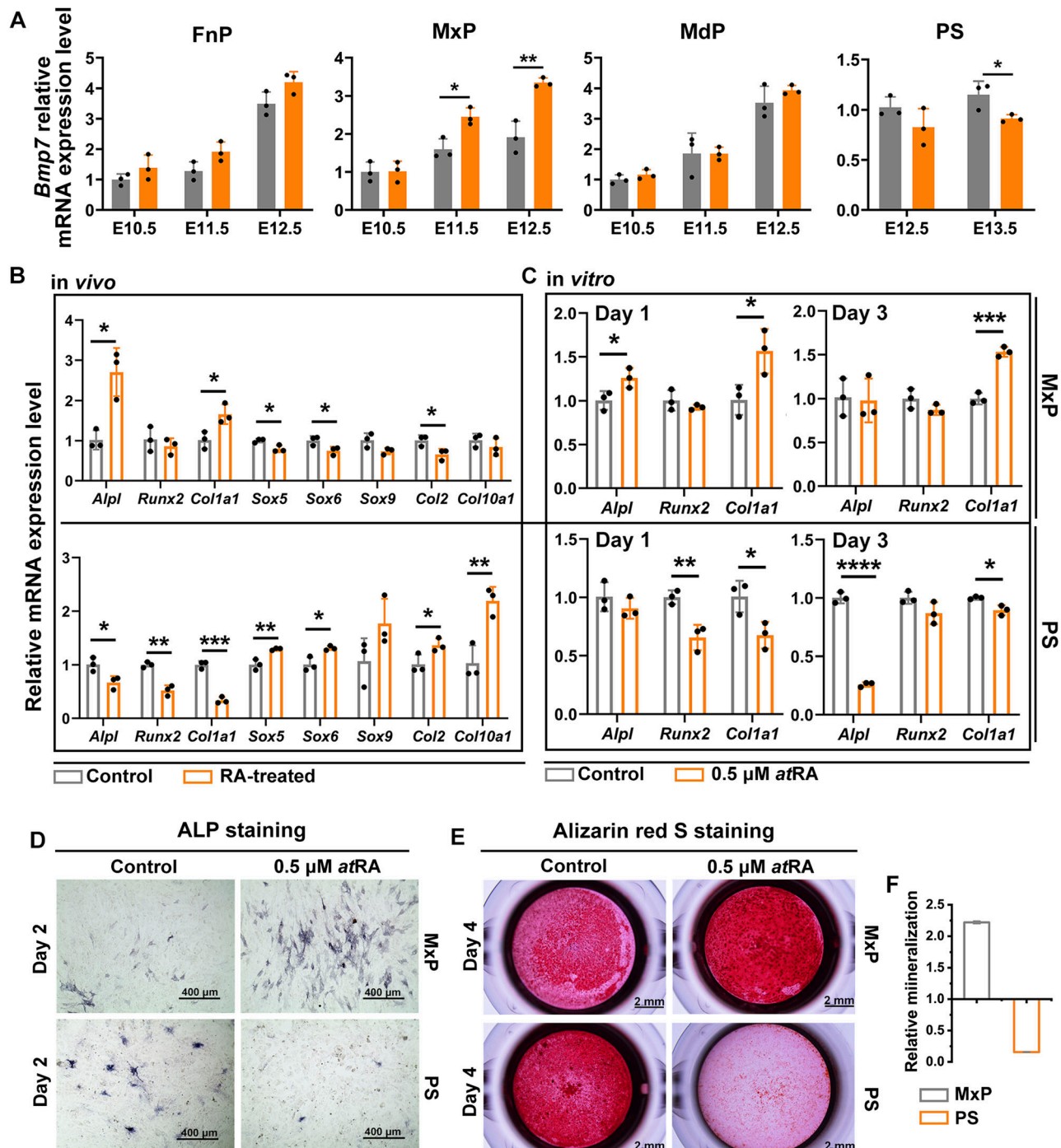

**Figure 7. RA induced the expression of osteogenic genes and enhanced osteogenic differentiation of mouse craniofacial mesenchymal cells.**
**(A)** qRT-PCR analysis showed the relative expression of *Bmp7* in frontonasal prominence, maxillary processes, mandibular processes and PS from E10.5–E13.5 mouse embryos of control and RA-treated mice. **(B)** qRT-PCR analysis of the expression of early (*Alpl*, *Runx2*), late *(Col1a1)* osteogenic differentiation genes, and cartilaginous markers (*Sox9*, *Sox5*, *Sox6*, *Col2*, and *Col10a1*) in maxillary-derived or PS-derived mesenchymal cells from E12.5 embryos of control and RA-treated mice. **(C)** qRT-PCR analysis of the expression of *Alpl*, *Runx2*, and *Col1a1* in maxillary-derived mesenchymal cells from E12.5 embryos of control mice induced with or without 0.5 $\mu$M atRA for 1 or 3 d. **(D)** Alkaline phosphatase staining of the maxillary-derived mesenchymal cells or PS-derived mesenchymal cells from E12.5 control mouse embryos cultured in osteogenic medium with or without 0.5 $\mu$M RA for 2 d. (Scale bar = 400 $\mu$m). **(E)** Alizarin red S staining of the maxillary-derived or PS-derived mesenchymal cells from E12.5 embryos of control mice cultured in osteogenic medium for 4 d with or without 0.5 $\mu$M RA in maxillary-derived mesenchymal cells and PS-derived mesenchymal cells. (Scale bar = 4 mm). **(F)** Mineralization quantitated from stained mineral deposits. Values were presented in means ± SD (n = 3). *$P < 0.05$, **$P < 0.01$, **$P < 0.01$, two-tailed, unpaired $t$ test.

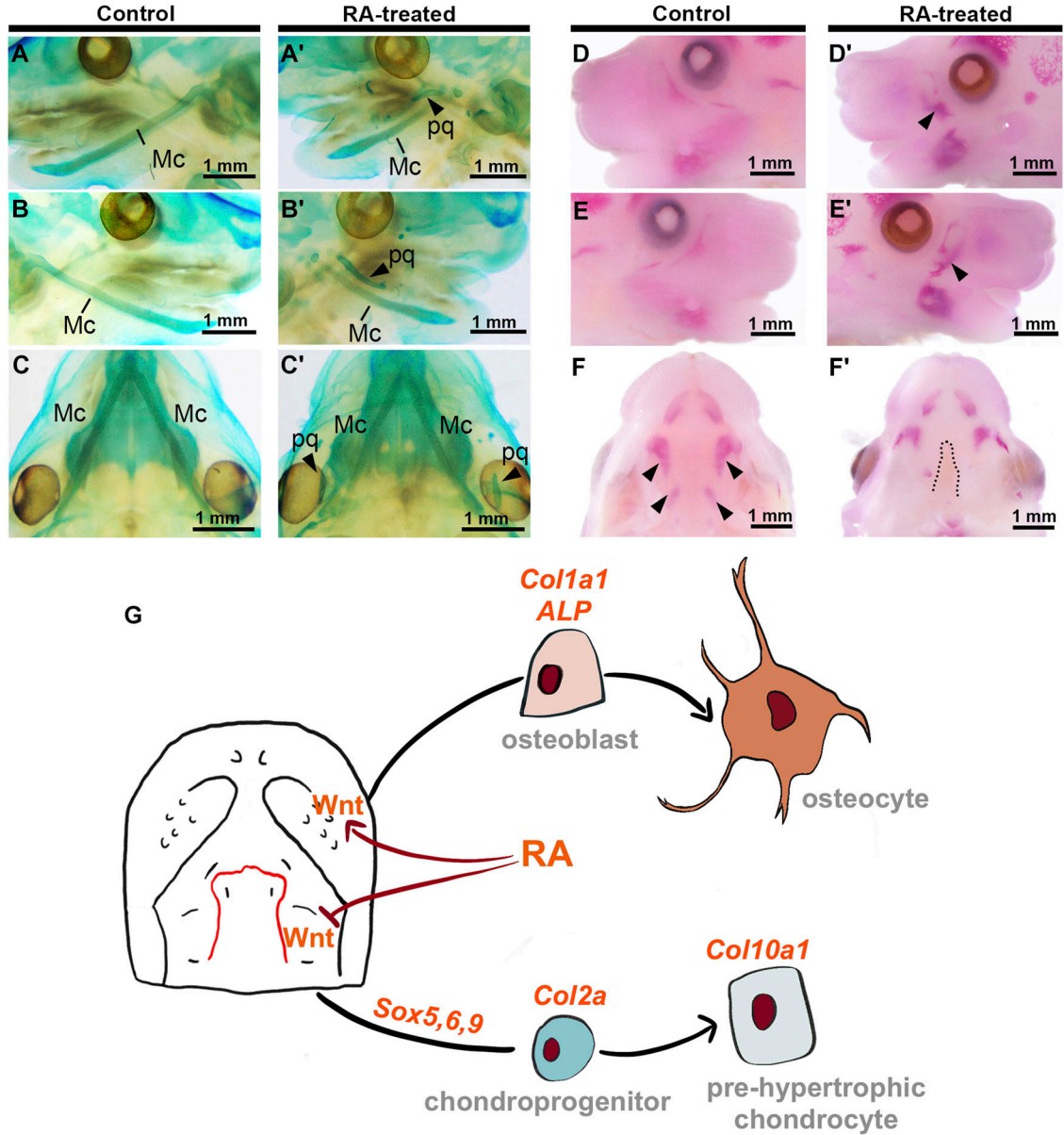

**Figure 8. Cartilage staining and osteogenesis staining of E15.5 embryos.**
**(A, A', B, B', C, C')** Lateral (A, B, A', B') and caudal (C, C') views of cartilage staining of E15.5 control (A, B, C) and RA-treated (A', B', C') embryos. Arrowheads indicate the aberrant palatoquadrates in an RA-treated embryo. **(D, D', E, E', F, F')** Lateral (D, D', E, E') and caudal (F, F') views of cartilage staining of E15.5 control (D, E, F) and RA-treated (D', E', F') embryos. Arrowheads indicate enhanced osteogenesis, dotted line showed failed fusion of PS in RA-treated mice. Scale bars: 1 mm; Meckel's cartilage, Mc; pq, palatoquadrates. **(G)** Schematic of gene expression and differentiation in response to RA-induction in facial prominences.

timings of formation and different genetic regulations (Jaruga et al, 2022). Although both genetic and environmental factors that affect palatogenesis have been investigated extensively, the molecular pathogenesis of upper lip cleft was poorly elucidated. Numerous studies have shown the pivotal roles of RA signaling in regulating the induction, migration, and differentiation of CNCCs, which give rise to most of the craniofacial ectomesenchymal derivatives (Williams & Bohnsack, 2019). The precise functions of RA activities are defined by the biological settings. For instance, diminishing RA expression at early developmental stage of E7.5 and E8.5 resulted in midfacial defects or cleft palate in mice, respectively (Wu et al,

2022). Excessive RA was shown to reduce sonic hedgehog in mouse craniofacial prominences whereas it was shown to induce Shh signaling in mouse suture–derived mesenchymal cells (James et al, 2010; Wang et al, 2019).

Proper RA signaling is also essential for osteogenic differentiation in craniofacial development, and excessive RA signaling cause enhanced osteogenesis due to the increased hedgehog and Bmps expression (James et al, 2010; Ferguson et al, 2018). The indispensable role of RA in frontonasal development has also been demonstrated in chicken embryo, in which duplication of upper beak/interorbital septum/prenasal cartilage/premaxilla,

and ectopic skeletal elements in palate were observed with excessive RA signaling in first pharyngeal arch (Abramyan & Richman, 2018). Although a number of studies attempt to elucidate the molecular pathogenesis of RA-induced cleft palate both in vivo and in vitro, the role of RA in regulating early facial development was barely addressed (Yao et al, 2011; Hu et al, 2013; Shu et al, 2018; Wang et al, 2019; Yoshioka et al, 2021a, 2021b). It was generally accepted that early synthesis of RA was restricted in the presomitic mesoderm diffused as far as rhombomere 3 (r3) and in the periocular mesenchyme, where RA functioned as an important morphogen to induce NCC migration (Williams & Bohnsack, 2019). However, the expression of *Rdh10* and *Aldh1a*s in the facial prominences identified in this study, along with the others, certainly indicated the endogenous RA synthesis in CNCCs and in epithelium. Therefore, the RA free local settings in the cranium and in the first pharyngeal arch were likely to be maintained by the expression of *Cyp26b1*, which might explain the discrepancies in the craniofacial manifestations observed in gain-of-function animal models of excessive RA signaling.

We identified a craniofacial dysmorphology–related regulatory gene network, consisting of miRNA, Wnt genes, Wnt-regulated transcription factors, and Bmp morphogens, in the mouse facial prominence subjected to early exposure of RA. In addition, we further characterized their dynamic change in expression at various timepoints during early–mid developmental stage. The overall up-regulated 12 Wnt-related genes were recognized as the potential targets of 14 down-regulated miRNAs identified in the RA miRNA-seq datasets of compared with that of the controls. The conserved miR-484 was recognized as the hub regulator targeting multiple Wnt-related genes including *Fzd5*, *Mapk9*, *Btrc*, and *Damm2*. The direct targeting of *Fzd5* by miR-484 was validated using luciferase assay in both 293T cells and MEPM cells. We further evaluated the expression of the Wnt receptor *Fzd5* and *Fzd9* and their cytoplasmic downstream effector *Dvl3* in discrete facial prominence between E10.5 and E12.5 and in the E12.5 and E13.5 PS. Similar pattern of RA-induced change of expression of *Fzd5* and *Dvl3* was observed in FnP and MxP, not in MdP, indicating the divergences in the regulation of developing program between MdP and FnP/MxP. Down-regulation of the expression of *Fzd5* and *Dvl3* was shown in E12.5 and E13.5 PS, in agreement with previous findings of the inhibition of Wnt-related gene expression in RA-treated MEPM cells by others (Hu et al, 2013). The differentially regulated expression of these Wnt-related genes was in accordance with the discrete pattern of RA-induced change of expression of miR-484 in these facial prominences. Further characterization of temporal- and tissue-specific transcriptional factors and enhancers in future research is going to help clarify the distinctively regulated expression of this microRNA in respond to RA signaling. The prominent-specific responses of Wnt signaling to RA induction were further verified by the TCF/LEF transcriptional activities determined using luciferase reporter assay. Two Wnt-regulated craniofacial defects-associated genes, *Foxn3* and *Itgb1*, were identified as the mediator of RA-induced osteogenic differentiation via up-regulation of *Bmp7* in FnP/MxP, demonstrated by the mRNA transcriptomic datasets and by the qRT–PCR analysis. In line with the differentially regulated Wnt activities in facial prominences, distinctive expression patterns of osteogenic and chondrogenic genes in response to RA exposure were also observed in MxP and PS, displayed by the ectopic chondrogenesis of palatoquadrates and the enhanced ossification in the proximal region of MxP/MdP. Our results support the different molecular regulation of lip formation and palatogenesis. A profound analysis of molecular pathways and morphological studies in future research will help understand the spatial–temporal cellular mechanisms underlining various forms of orofacial clefts caused by different genetic/environmental risk factors.

## Materials and Methods

### Reagents

All-trans retinoic acid (*at*RA) (Cat. No. R2625), corn oil (Cat. No.C8267), Alcian Blue 8GX (Cat. No. A5268), alizarin red S (Cat. No. A5533), DMSO, 4% PFA, ethanol, acetic acid, benzyl alcohol, benzyl benzoate, and Hoechst 33342 (Cat. No. B2261) were from Sigma-Aldrich Co. (Cat. No.C11965500BT; DMEM), (Cat.No.10099-141; FBS), and 0.25% trypsin (Cat.No.25200-072) used in cell culture and TRIzol reagent (Cat.No.15596018) used in RNA extracting were purchased from Invitrogen (Invitrogen). Gelatin (Cat. No. 07903) was from STEMCELL Technologies, (Cat. No. SV30010; penicillin/streptomycin) was from Hyclone. NBT/BCIP stock solution (Cat. No. 11681451001) used for alkaline phosphatase staining was from Roche. PrimeSTAR DNA polymerase, *Sac I*, *Xba* I, *BamHI*, T4 DNA ligase were purchased from Takara Bio, Inc. The double-strand miR-484 mimics (5′-UCAGGCUCAGUCCCCUCCCGAU-3′) and the scramble RNAs (5′-UUCUCCGAACGUGUCACGUTT-3′) were synthesized by GenePharma.

### Animals

All procedures conducted in this study involving animals were approved by the Institutional Animal Care and Use Committee of Jinan University (Approval No. 20220222-58) and were in strict accordance with the National Institutes of Health (NIH) Guide for the Care and Use of Laboratory Animals. The inbred C57BL/6J mice (age, 10–12 wk; weight, 25–30*g*) were obtained from The Experimental Animal Center of Guangdong (Guangzhou, China). All mice were housed in a pathogen-free facility at the Laboratory Animal Management Center of Jinan University at 22°C with 12-h light/12-h dark cycle and were provided access to food and water ad libitum. Mice were mated overnight, and embryos were considered E0.5 when a vaginal plug was detected in the morning. The embryonic developmental stages were further verified by examining of the craniofacial features after dissecting, craniofacial anatomy of embryos was referred to FaceBase database (https://www.facebase.org/mouseanatomy/). The pregnant mice were administered with one dose (25 mg/kg, diluted 1/10 in corn oil) of *at*RA (25 mg/ml in DMSO) by oral gavage for three consecutive days at E8.5, E9.5, and E10.5, whereas the control mice received vehicle only (Wang et al, 2019). Mice were euthanized by $CO_2$ gas, and the embryos at the indicated developmental stages were collected by cervical dislocation according to the American Veterinary Medical Association Guidelines.

## Sample collection and RNA extraction

The embryonic facial prominences including medial and lateral FnP, maxillary and mandibular prominences were isolated in icy-cold PBS, as described in previous publication (Feng et al, 2009). All tissue samples were flash frozen in liquid nitrogen and then stored at –80°C before use. One embryo from each litter was collected for mRNA and miRNA sequencing. The craniofacial malformations of RA-treated embryos were verified before sampling. Total mRNA and miRNA were extracted using miRNeasy Mini Kit (Cat.No.217004; QIAGEN, GmBH) and mirVana miRNA Isolation Kit (Cat.No.AM1561; Ambion), respectively, according to the manufacturer's instructions. The concentrations of RNA samples were determined from the absorbances at 260 nm using a NanoDrop ND-2000 spectropho-tometer (Thermo Fisher Scientific). The RNA integrity numbers (RIN) were quantified on an Agilent 2100 Bioanalyzer (Agilent technol-ogies) and only samples with RIN ≥7 were processed for subsequent sequencing.

## mRNA-seq and miRNA-seq

For mRNA sequencing, poly(A)-selected mRNA was enriched and rRNA was removed from ~1 $\mu$g total RNA using the TruSeq RNA Sample Prep V2 Kit (Cat.No.RS-122-2001; Illumina) before con-structing the library for mRNA-seq. In brief, poly-A mRNA was extracted using oligo-dT–coupled magnetic beads and was sub-sequently fragmented. Fragmented RNA was reversed transcribed into first-strand cDNA, followed by second-strand cDNA tran-scription. The barcodes and adapters were then added to the cDNA fragments to generate the cDNA libraries with an average fragment size of 466 bp. The libraries were quantified and validated using a Qubit 2.0 Fluorometer (Life Technologies) and Agilent 2400 bio-analyzer (Agilent Technologies) system. These libraries were used for sequencing to get an average read depth of 40–50 million reads per sample on Illumina NovaSeq 6000 sequencing platform (Illu-mina) for 150 bp paired-end reads (PE150).

For miRNA sequencing, miRNA was isolated from 10 ng total RNA using mirVana miRNA Isolation Kit. The standard miRNA sequencing libraries were prepared using NEBNext Multiplex Small RNA Library Prep Set for Illumina (Cat.No. E7580S; New England Biolabs) and were sequenced at an average read depth of 18–20 million reads per sample on an Illumina NovaSeq 6000 system using PE150 model (Shanghai Biotechnology Corporation).

Raw data of sequencing are available in GEO dataset GSE238064, including individual data series GSE238062 and GSE238063 for RNA-seq and miRNA-seq samples, respectively.

## mRNA and miRNA sequencing data processing

Quality control of paired-end FASTQ files (raw data) that came from sequencing pipelines were performed using FastQC (https://www.bioinformatics.babraham.ac.uk/projects/fastqc/). Then, trim-med off the barcodes and adapters, and removed sequences that are of low quality or too short (remove the reads shorter than 25 bp) using Seqtk to obtain clean reads. The remaining clean reads of mRNA-seq data were mapped to the mouse reference genome mm (10) (GRCm38.p4) using Hisat2 (version 2.0.4) for genome annotation

annotations and miRNA-seq data were aligned to the miRNAs in miRbase20.0 (http://www.mirbase.org/) using Bowtie2 software (version 2.4.4) (Langmead et al, 2009; Kozomara & Griffiths-Jones, 2014; Kim et al, 2015). The gene expression levels were quantified with StringTie (version 1.3.0) using only uniquely mapped reads (Pertea et al, 2015; Pertea et al, 2016). Given the different lengths of genes and the amount of sequencing data for each sample, the read counts were normalized to sequencing depth and the length of the gene at the same time, to obtain values of fragments per kilobase of transcript per million mapped reads. Read counts of miRNA were normalized to gene length first, and then normalized for sequencing depth, final, to get values of transcripts per million (Mortazavi et al, 2008). Unannotated miRNAs were evaluated for potential novel miRNA candidates using miRCat pipeline in the UEA sRNA workbench (version 4.4) according to miRNA precursor hairpins (Stocks et al, 2012, 2018).

## Identification of differentially expressed mRNAs (DEGs) and miRNAs (DEMs)

Differential expression analysis between treatment group and control group was determined using "glmQLFit" function of the edgeR package in R (version 3.6.3) (Robinson et al, 2010; Ziats & Rennert, 2014). Gene expression data were included only if their expression were found in at least four samples with count-per-million values ≥ 10, filtered using the "filterByExpr" function, and normalized to the trimmed mean of M values. The differentially expressed multiples (fold change) were calculated for DEGs and DEMs, with $P$-value < 0.05 considered statistically significant. The volcano plots of DEGs and DEMs were constructed by plotting $\log_{10}$-transformed $P$-value on the y-axis and $\log_2$-transformed fold change on the x-axis using "ggplot2" package of R (version 3.6.3). The heatmaps of DEGs and DEMs were generated using "pheatmap" package of R (version 3.6.3), and hierarchically clustered according to Pearson correlation and average linkage. The principal com-ponent analysis of the normalized gene expression data was performed using the "prcomp" function in R (version 3.6.3).

## Enrichment analysis for mRNA-seq datasets

GO-BP functional enrichment analysis and KEGG pathway enrich-ment analysis of DEGs were performed using DAVID (v2022q2, https://david.ncifcrf.gov/) based on total genes in the genome as a population background, with the overlapping/redundant terms identified and removed (Huang et al, 2007; Hooper et al, 2017). Enrichment analysis was subjected to Fisher's exact test, with false discovery rate adjusted $P$-value < 0.05 considered as significant.

## Construction of the miRNA–gene interaction network and identification RA-induced cleft-associated genes regulated by Wnt

miRNAs potentially regulating the up-regulated Wnt-related genes were predicted by the online tool of TargetScan 8.0 (https://www.targetscan.org/) and miRWalk (version 3, http://mirwalk.umm.uni-heidelberg.de/). miRNAs recognized by both programs in RA-induced down-regulated DEMs were identified by

VennDiagram package (version 1.7.3). The miRNA-mRNA regulatory network was visualized using Cytoscape (version3.9.1). The hub miRNAs were identified by MCC score calculated using cytoHubba plugin (version 0.1). Wnt-regulated genes containing TCF/LEF motif were identified using hTFtarget database (http://bioinfo.life.hust.edu.cn/hTFtarget/), in which human transcription factors (TFs) were curated from Chip-Seq experiments in different cell lines, tissues, and cells (Zhang et al, 2020). A list of cleft-associated genes was compiled by searching Mouse Genome Informatics and comprehensive automatically mined database of human diseases (Malacards) using the following key words: cleft lip, cleft palate, orofacial cleft, craniofacial phenotype, or craniofacial malformations. Redundant terms were filtered manually.

### Real-time quantitative PCR

1 $\mu$g of total RNA, extracted using TRIzol, was reverse transcribed to cDNA using PrimeScript RT Reagent Kit with gDNA Eraser (Cat. No. RR047A; Takara), according to the manufacturer's instructions. qRT–PCR were performed using TB Green Premix Ex TaqII (Cat. No. RR820A; Takara) on an Applied Biosystems Quanstudio3 (Applied Biosystems). The quantification of target gene mRNA expression was normalized to the expression of *GAPDH*, and the relative expression levels were calculated using $2^{-\Delta\Delta Ct}$ method (Livak & Schmittgen, 2001). miR-484 was reversed transcribed using a stem-loop primer and amplified with a pair of primers. Relative quantification of miR-484 expression was analyzed using U6 as an internal control. All primers used in this study containing all pairs of qRT-PCR primers were designed with Oligo7 (version 7.5.6) software and are listed in Table S10.

### Cell culture and TOP/FOP-flash assays

The primary mouse embryonic craniofacial mesenchymal cells from MxP and the MEPM cells from PS were isolated and cultured from E12.5 mouse embryo (Yoshioka et al, 2022). MxP and PS were dissected from E12.5 mouse embryos in cold PBS, followed by digesting in 0.25% trypsin for 30 min at 37°C in a $CO_2$ incubator. The cells were then centrifuged at 300$g$ for 5 min and the pellet were resuspended cells in 2 ml culture medium. The cells were plated onto six-well plates precoated with 0.1% gelatin (Cat.No.07903; STEMCELL Technologies). Primary mesenchymal cells were maintained in DMEM supplemented with 10% FBS, 1% penicillin/streptomycin, and 2 mM L-glutamine at 37°C in a $CO_2$ incubator with 5% $CO_2$. The Wnt activities were assessed using a TOP/FOP reporter system and dual-luciferase reporter kit (Cat.No.2920; Promega). The cells were co-transfected with pRL-TK plasmid (60 ng/well) and either TOP-flash plasmid or FOP-flash plasmid (600 ng/well) using Lipofectamine 2000 reagent (Cat.No. 11668027; Invitrogen) in 24-well petri dishes. After 48 h, the luciferase activities were detected using dual-luciferase reporter kit and the fluorescence intensities were measured using a luminometer (CLARIOstar Plus, BMG Labtech). The ratios of firefly/Renilla luciferase intensities were calculated.

### Dual-luciferase reporter gene assays

515 bp fragment of *Fzd5* 3'UTR containing the clustered three miR-484 target sites predicted by TargetScan was amplified from the cDNA of MEPM cells (forward primer: 5'- ATAAA-GAGCTCCCCTGAACTCAAGGTTTAAAG-3'; reverse primer, 5'-ATAAATCTA-GAGGGAACTACGCTTTTTTCC-3'). Mut-*Fzd5*-3'UTR with sense mutations at the seed sequences of all the three predicted miRNA: mRNA interaction sites were synthesized by Sangon Biotech. Both fragments of WT-*Fzd5*-3'UTR or Mut-*Fzd5*-3'UTR were cloned into *Sac* I/*Xba* I site of pmirGLO dual-luciferse miRNA target expression vector (Promega), respectively. Both constructs were co-transfected with either miR-484 mimics or scrambled control RNA, respectively, into primary MEPM cells using Lipofectamine 2000 reagent (Invitrogen). After 48 h, cells were harvested for dual-luciferase reporter 1,000 assay (Cat. No. E2920; Promega). The luciferase activities were measured using a luminometer (CLAR-IOstar Plus, BMG Labtech) and were presented as ratios of firefly/Renilla.

### Promoter analysis

The sequence of 1,442 bp promoter–enhancer region of miR-484 (Hu et al, 2019) was analyzed using ALGGEN-PROMO (version 3.0.2) (https://alggen.lsi.upc.es/) to identify putative cis-regulatory elements. 500 bp fragment of the proximal promoter region, from −495 to +5, containing the two putative RAREs was amplified from mouse genomic DNA using primer pairs (forward primer: 5'- ATAAAATC-TAGAGGCGCCAACAGTGACAGTAGT-3'; reverse primer, 5'- ATAAAAG-GATCCAGGCTGCAGGGCCGCGA-3') and was subsequently cloned into the *Xba* I and *Bam*HI sites of the vector pEZX-FR01 (GeneCopoeia Inc.). The fragment of the same promoter sequence containing mutation at the second RARE was synthesized by Sangon Biotech and was cloned into the same sites of pEZX-FR01. The promoter activities were evaluated in 293T cells induced with or without 1 $\mu$M *at*RA for 48 h using the dual-luciferase reporter assay system (Promega), according to the manufacturer's instructions.

### Hoechst staining

Whole-mount Hoechst staining was used to intensify the imaging of embryonic tissues for morphological analysis (Sandell et al, 2012). Embryos of various developmental stages were fixed in 4% PFA at 4°C overnight, followed by overnight incubation in 1 $\mu$g/ml Hoechst 33342 trihydrochloride trihydrate (Invitrogen). Micrographs of mouse embryos were captured using a Nikon stereomicroscope (SMZ18; Chiyoda-ku).

### Bone and cartilage staining

Cartilage staining of E15.5 embryos was conducted as previously described (Nagy et al, 2009). The E15.5 embryos were fixed in Bouin's fixative for 2 h, followed by 24 h treatment at RT in ammonium hydroxide: ethanol solution (0.1% ammonium hydroxide, 70% ethanol), with exchanging for fresh solution several times until the embryos turned white. The embryos were then soaked twice in 5% acetic acid for 1 h and stained in Alcian Blue (0.05% Alcian Blue [wt/

vol] in 5% acetic acid [vol/vol]) for 2 h. After washing two times in 5% acetic acid twice for 1 h, the samples were then cleared in methanol (two times for 1 h) and stored in BABB (mix benzyl alcohol:benzyl benzoate at a ratio of 1:2). Bone staining of E15.5 embryos was performed after 2 d fixation in 100% ethanol. The embryos were then transferred to 4% PFA overnight, followed by two times wash in water before soaking in 0.5% potassium hydroxide for 6 h. The embryo skeletons were stained in alizarin red (50 µg/ml in 0.5% potassium hydroxide) overnight and washed with 0.5% potassium hydroxide for 6 h to remove the excess stain. The embryos were then placed in 0.2% potassium hydroxide/20% glycerol until clear and then stored in 100% glycerol. Images were captured under a Nikon stereomicroscope (SMZ18; Chiyoda-ku) using a Nikon's DS-Fi3 microscope camera and processed with NIS-Elements software.

### Analysis of osteogenic capacity of mouse embryonic mesenchymal cells in vitro

Confluent mouse embryonic mesenchymal cells from E12.5 MxP and from E12.5 PS on 24-well plates were induced in osteogenic medium composed of DMEM, 10% FBS, $10^{-8}$ M dexamethasone, 50 mg/ml of penicillin/streptomycin, 5 mmol/l of $KH_2PO_4$, and 50 mg/ml of L-ascorbic acid with or without 0.5 µM *at*RA. The expression of biomarkers of osteogenic differentiation including *Alpl*, and late osteogenic gene, *Col1a1* was determined using qRT–PCR with gene-specific primers (Table S10). The alkaline phosphatase activities in fixed (in 4% PFA for 30 min at RT) cultured cells were also visualized by colorimetric reaction using NBT/BCIP stock solution (Cat. No. 11681451001; Roche), according to the manufacturer's instructions. The cells were also fixed in 95% ethanol for mineralization assay. The cells were stained with 1% alizarin red S for 20 min at RT, then washed thoroughly with PBS. The images were captured using a Nikon stereomicroscope (SMZ18; Chiyoda-ku). After imaging, 600 µl 10% (vol/vol) acetic acid was added into each well containing cells stained with alizarin red and incubated at RT for 30 min with shaking. The samples were then collected with a cell scraper and transferred to a microcentrifuge tube. After a brief vortex mixing, the samples were incubated at 85°C for 10 min, followed by 5 min incubation on ice. After centrifuging at 20,000*g* for 15 min, 500 µl supernatant was collected, and 200 µl of 10% (vol/vol) ammonium hydroxide was added to each sample. The absorbance at 405 nm were read in 96-well plate using a luminometer (CLARIOstar Plus, BMG Labtech), and the amount of alizarin red was quantified using a standard curve and expressed as mM.

### Statistical analysis

At least three independent experiments were performed for all assays, and the results were presented as mean ± SD, with the result of each individual experiment indicated. Statistical analysis was performed by GraphPad Prism 8.0 software (GraphPad). Comparisons between two groups were subjected to *t* tests (unpaired, two-tailed) whereas one-way ANOVAs followed by post hoc test were used for multiple comparisons. Differences were considered statistically significant at *P* < 0.05.

## Data Availability

The datasets used and analyzed during the current study are available from the corresponding author on reasonable request. The raw data are available via GEO (http://www.ncbi.nlm.nih.gov/geo) with the accession number GSE238064, including individual data series GSE238062 and GSE238063 for RNA-seq and miRNA-seq samples, respectively.

## Supplementary Information

## Acknowledgements

This work was sponsored by the Natural Science Foundation of Guangdong (#2023A1515010293).

### Author Contributions

C Song: data curation, formal analysis, visualization, and methodology.
T Li: validation and investigation.
C Zhang and S Li: investigation.
S Lu: data curation.
Y Zou: conceptualization and writing—original draft and project administration.

### Conflict of Interest Statement

The authors declare that they have no conflict of interest.

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
