## [Reviewer comments · Life Science Alliance]

Life Science Alliance

RA-induced prominence-specific response resulted in distinctive regulation of Wnt and osteogenesis

Chao Song, Ting Li, Chunlei Zhang, Shufang Li, Songhui Lu, and Yi Zou

DOI: <https://doi.org/10.26508/lsa.202302013>

Corresponding author(s): Yi Zou, Jinan University

Review Timeline:

Submission Date:	2023-02-25
Editorial Decision:	2023-03-27
Revision Received:	2023-06-21
Editorial Decision:	2023-07-16
Revision Received:	2023-07-24
Accepted:	2023-07-27

Scientific Editor: Novella Guidi

Transaction Report:

March 27, 2023

Re: Life Science Alliance manuscript #LSA-2023-02013-T

Dr. Yi Zou
Jinan University
The Key Laboratory of Virology of Guangzhou
Jinan University, No.601, West Huangpu Avenue , Guangzhou, Guangdong, China.
Guangzhou 510632
China

Dear Dr. Zou,

Thank you for submitting your manuscript entitled "Prominence-specific response to early RA exposure resulted in distinctive regulation of Wnt signaling and osteogenesis" to Life Science Alliance. The manuscript was assessed by expert reviewers, whose comments are appended to this letter. We invite you to submit a revised manuscript addressing the Reviewer comments.

Thank you for this interesting contribution to Life Science Alliance. We are looking forward to receiving your revised manuscript.

Sincerely,

B. MANUSCRIPT ORGANIZATION AND FORMATTING:

Reviewer #1 (Comments to the Authors (Required)):

The manuscript by Song et al describes an analysis of craniofacial development in midgestation mouse embryos from dams treated with 3 dose of all-trans retinoic acid. The manuscript focuses primarily on an analysis of gene and miRNA expression data obtained from dissected craniofacial prominence, with some validation analyses and corresponding assessment of phenotypic changes.

Overall, there are some positives to the manuscript, including the clear presentation of gene expression data in the figures and the clearly described and detailed methods. There have been many studies conducted in mice and other species looking at the role of retinoic acid in craniofacial development, which the authors reasonably acknowledge, and indeed many of the results presented are confirmatory in this regard. That said, the authors do present some interesting new findings in the gene expression analysis that would be of interest to those in the field. Even so, there are a number of significant concerns that must be addressed.

1. With respect to writing style and presentation, parts of the manuscript are well written, with good English (eg. Methods and much of the Discussion) but other parts are riddled with clumsy statements, and problems with tense and grammar, and a word duplication. These are too many to list here. An example is the sentence in the Abstract beginning with "While inhibited the Wnt activities...".
2. The Introduction is broad and tries to cover too much. While an introduction to facial development is warranted, this should be more brief covering the parts only relevant to the manuscript, with more focus of just Wnt signaling, rather than highlight a multitude of other signaling pathways and gene names which do not come up later. It should also not summarize all the results, just the main broad findings.
3. The authors seem to be emphasizing the relevance to cleft lip early in the manuscript yet the experimental design is aimed at looking at gene expression at least 2 days after the lip has formed. The manuscript also aims to tie some of the gene expression changes to secondary palate development, which is more appropriate if considering the very early stages of palatal shelf growth. But then they transition later to talking about osteogenesis (which has not significantly initiated at the time the gene expression analysis is done) and then in the Discussion the authors start talking about cartilage development. While they present some data related to each of these aspects, there are large assumptions made linking them all based on the data they present. While there is a reasonable case to be made for these linkages based on the fullness of the existing literature, to do so based on just the presented data is a stretch and makes the manuscript seem somewhat unfocused. Furthermore, the appearance of palatoquadrates, which has previously been reported, may be associated with cartilage and bone development but the appearance indicates an earlier patterning defect which precedes skeletogenesis and therefore the latter may be secondary to the influence of RA.
4. Much of the important message of the manuscript is built upon the gene expression data. While the authors have appropriately included data from 4 WT and 4 treated embryos, there is no mention of whether each group of 4 are matched for gender. The methods do not indicate that sexing of embryos was performed. Inclusion of mixed sexes in RNAseq data can have a profound impact on the analysis of differential gene expression - for example if the WT group had 1 female and 3 males vs the treated group having 1 male and 3 females. The impact is not just restricted to X-linked genes. This potentially represents a significant concern as to the overall robustness of the data. There of course will still be real differences that can be observed even without sex-matching (like some of the examples reported and validated - eg Foxn3) but it can be severely limited.
5. With respect to the two genes (Foxd3 and Itgb1), that are validated by the authors, they mention these genes are associated with craniofacial malformations but do not say what type of malformations are associated with these genes. It needs to be stated. Is it not because it is not cleft lip/palate as per the emphasis of the manuscript?
6. Some references are needed for certain statements, such as 'excessive intake of vitamin A' on line 79. Presumably the authors are referring to human studies?? And this is dietary supplementation? If so, such supplementation is associated with an increased risk of various malformations. It is only the teratogenic studies in humans that provide the strongest tie of RA excess to craniofacial malformations. Dietary excess is implicated but the level of intake required is ill-defined.

7. Given the RNAseq analysis was performed at E12.5 and the biological processes discussed (lip formation and skeletogenesis) are not occurring at this time (ie. it is NOT a converging point as stated on line 129), the rationale for assuming all differentially expressed Wnt related genes are regulated by miRNAs seems a stretch. Clearly the authors have found evidence that some are, which is nice. But the way this is presented should be more appropriately written and justified.

8. The authors early on (line 91) state that multiple doses of atRA were used to induce full penetrance CL/P. However, in the data presented (Fig 1), only one of the shown embryos has CL/P. One has CPO, and one purportedly has midfacial hypoplasia although this is not apparent from the image nor is there quantitative data to support this. For statements on penetrance, it is necessary to catalogue and present the incidence of all features in all embryos. It is not even mentioned how many embryos have been examined in this study. What percent have CL/P, CPO, CL, midfacial hypoplasia, mandibular clefts, neural tube defects, eye defects, etc? How do these compare to prior studies? (this goes to the robustness of the treatment regimen). The statement 'various incidences in all embryos' is incorrect as 'incidence refers to occurrence in the entire group not in specific embryos. It is better to say 'variable phenotypes in all embryos' but again this needs to be documented.

9. mRNA-seq and miRNA-seq 'databases' - 'databases' is the wrong word here. Datasets is more accurate.

10. For the analysis of the RNA sequencing datasets, stating that 23173 (66.7%) were expressed in at least 4 samples is potentially misleading. Such statements are only useful if it relates to the 4 samples in that group (eg. all 4 WT sample or in all 4 treated embryos). Same applies to the miRNA seq statements. What % are in all 4 of each group? This is important to gauge reproducibility.

11. One line 168, the authors refer to Rxr but not which isoform(s) in the MdP despite stating isoform information for the MxP.

12. Be more specific about the functions of the genes involved in Wnt signaling that are upregulated. Are the positive effectors, downstream targets or inhibitors? (line 188). A latter statement (lines 225-227) makes it unclear, thus necessitating clarity earlier.

13. Again, the use of cleft lip as a keyword may not be the most appropriate given the analyses were done two days after fusion of the lip region when some of the known CL/P genes may be turned off.

14. Fig 1 needs to be accompanied by a summary table presenting the phenotypic variation and incidence of each presentation (eg. CL/P). Panel F is meant to be wildtype yet shows facial anomalies (hypoplasia), eye malformations, vascular anomalies and a neural tube defects!? Whereas the treated embryo labeled as F' appears normal. An 'n' value for the number of embryos examined at each age needs to be mentioned in the legend and to address significance of all conclusions. The legend also does not mention any phenotype at all in B'. In E' the phenotype appears restricted to the mandible and therefore not CLP. The scale bar for the left side images (eg. A) is stated as 1mm. However the length of the bar is significantly larger in A' suggesting the treated embryo is notably smaller than wildtype. Is this true? Or is the scalebar wrong? Is this an outlier or is it statistically significant? Is this size difference consistent at all stages of development? Panel H should be in the bottom right corner not above F and G.

15. In Fig 8 legend, it notes the abbreviation "PS" as palatoquadrate, which it is not - should be palatal shelf.

Reviewer #2 (Comments to the Authors (Required)):

In this article, authors investigated the spatiotemporal effect of excess RA exposure on regulating early development of facial prominences. The data showed that early exposure of RA inhibited the Wnt activities in E12.5/E13.5 mouse palatal shelves, but induced Wnt signaling and Wnt-related gene expression in mouse embryonic Frontonasal/Maxillary processes. A conserved regulatory network of miR-484-Fzd5 was identified, which may play critical roles in RA regulated craniofacial development. Foxn3 and Itgb1 can be induced by early RA-exposure. Besides, expression of BMP7 can also be affected by RA-exposure. This study is very interesting and may expand our knowledge about the role of RA in the development. There are several comments for this article.

1. There are some typos need to be corrected.

2. Author found the regulatory network miR-484-Fzd5. What's the relationship between miR484-Fzd5 and Foxn3 or Itgb1 in excessive RA-exposure? More in vitro data should be provided to confirm the results of bioinformatic analysis.

3. What's the relationship between miR484-Fzd5 and BMP7, or Foxn3 / Itgb1 and BMP7 in excessive RA-exposure? More in vitro data should be provided to demonstrate the possible intrinsic connection.

Dear editor,

The reviewer's comments are highly appreciated and have been carefully considered to help us make substantial improvements to our manuscript. A point-to-point response to the reviewer's comments is presented below. Besides all the corrections we made according to reviewers' suggestions, we further clarified the potential molecular mechanisms of the tissue-specific response to RA induction by characterizing the cis-regulatory element of RARE in the promoter of *miR-484*, the hub miR identified in our study and verified its tissue-specific regulation by RA. We believed that these results substantially improved the understanding of the underpinning molecular mechanisms and the quality of the manuscript.

Reviewer #1 (Comments to the Authors (Required)):

1. With respect to writing style and presentation, parts of the manuscript are well written, with good English (eg. Methods and much of the Discussion) but other parts are riddled with clumsy statements, and problems with tense and grammar, and a word duplication. These are too many to list here. An example is the sentence in the Abstract beginning with "While inhibited the Wnt activities...".

Response: The English writing has been thoroughly checked by the authors as well as using Microsoft Word Editor. All corrections were highlighted in the manuscript.

2. The Introduction is broad and tries to cover too much. While an introduction to facial development is warranted, this should be more brief covering the parts only relevant to the manuscript, with more focus of just Wnt signaling, rather than highlight a multitude of other signaling pathways and gene names which do not come up later. It should also not summarize all the results, just the main broad findings.

Response: The elaboration of early signaling pathways (line 61-66, line 73-78, in the original manuscript) that were not relevant to this study was eliminated to keep the "introduction" focused. The last paragraph of "introduction" was rephrased, as highlighted, to only present the main findings of this research.

3. The authors seem to be emphasizing the relevance to cleft lip early in the manuscript yet the experimental design is aimed at looking at gene expression at least 2 days after the lip has formed. The manuscript also aims to tie some of the gene expression changes to secondary palate development, which is more appropriate if considering the very early stages of palatal shelf growth. But then they transition later to talking about osteogenesis (which has not significantly initiated at the time the gene expression analysis is done) and then in the Discussion the authors start talking about cartilage development. While they present some data related to each of these aspects, there are large assumptions made linking them all based on the data they present. While there is a reasonable case to be made for these linkages based on the fullness of the existing

literature, to do so based on just the presented data is a stretch and makes the manuscript seem somewhat unfocused. Furthermore, the appearance of palatoquadrates, which has previously been reported, may be associated with cartilage and bone development but the appearance indicates an earlier patterning defect which precedes skeletogenesis and therefore the latter may be secondary to the influence of RA.

Response: The RA exposure induced full penetrance of CP, as described in previous study (Wang et al.). Cleft lip was not observed in all RA-induced embryos, although other forms of craniofacial malformations were presented. We focused on the change of gene expression in craniofacial prominences after early exposure to excessive RA. We rephrased the first sentence of the last paragraph of “introduction” (CLP was substituted with craniofacial malformation in here as well as in the subsequent relevant texts) to make it more appropriate, as highlighted in the revised manuscript.

E12.5 mouse embryos were selected for sequencing because this is the turning point from the completion of facial development to the initiation of palatal development. Also, at this stage, all prominences including FnP/MxP/MdP/PS can be easily isolated with good reproducibility for following assays. In addition, the skeletal progenitors’ condensation and differentiation were established at E12.5 in the mesenchyme of facial prominence. Therefore, the stretchiness of our data was a reasonable reflection of the nature of this developmental stage. However, we also realized the limitations of the RNA-Seq data obtained from the single time point. And, that is why we verified our initial findings derived from RNA-Seq in various developmental stages from E10.5 to E15.5 in discrete embryonic tissues using multiple relevant functional assays.

Yes, I agree with the reviewer that the appearance of palatoquadrates is more a consequence of change in pattern of differentiation. The use of “skeletogenesis” was inappropriate and was amended/heighted in the relevant texts.

4. Much of the important message of the manuscript is built upon the gene expression data. While the authors have appropriately included data from 4 WT and 4 treated embryos, there is no mention of whether each group of 4 are matched for gender. The methods do not indicate that sexing of embryos was performed. Inclusion of mixed sexes in RNAseq data can have a profound impact on the analysis of differential gene expression - for example if the WT group had 1 female and 3 males vs the treated group having 1 male and 3 females. The impact is not just restricted to X-linked genes. This potentially represents a significant concern as to the overall robustness of the data. There of course will still be real differences that can be observed even without sex-matching (like some of the examples reported and validated - eg Foxn3) but it can be severely limited.

Response: Yes, we also noticed that the sex differences could add complexity to transcriptome profile. It would be better if the sequencing data had been obtained from sex-matched samples. However, to the best of our knowledge, sex-specific roles of Wnt signaling in regulating early craniofacial development have not been reported. In addition, all initial gene expression data have been validated in at least three biological replicas at each different developmental stage.

5. With respect to the two genes (Foxd3 and Itgb1), that are validated by the authors, they mention these genes are associated with craniofacial malformations but do not say what type of malformations are associated with these genes. It needs to be stated. Is it not because it is not cleft lip/palate as per the emphasis of the manuscript?

Response: According to the information in MGI and the indicated reference in our manuscript, loss-of-function mutation of Itgb1 in mice resulted in several craniofacial malformations including cleft palate. Limited information on Foxn3 in regulating craniofacial development was available. Null mutation of Foxn3 resulted in severe craniofacial anomalies in mice including malformation in frontal, parietal, occipital, mandible bones as well as palatal bones. We briefly described the phenotypic defects in our original manuscript (line 281-282). Description of craniofacial phenotypes with more details was included and highlighted in our revised manuscript (line 297-301).

6. Some references are needed for certain statements, such as 'excessive intake of vitamin A' on line 79. Presumably the authors are referring to human studies?? And this is dietary supplementation? If so, such supplementation is associated with an increased risk of various malformations. It is only the teratogenic studies in humans that provide the strongest tie of RA excess to craniofacial malformations. Dietary excess is implicated but the level of intake required is ill-defined.

Response: Excessive intake of vitamin A in human can occur in the form of dietary supplementation as well as in the form of isotretinoin, a retinoid that is commonly prescribed for recalcitrant cystic acne. References for this statement were added in the revised manuscript (line 77-78).

7. Given the RNAseq analysis was performed at E12.5 and the biological processes discussed (lip formation and skeletogenesis) are not occurring at this time (ie. it is NOT a converging point as stated on line 129), the rationale for assuming all differentially expressed Wnt related genes are regulated by miRNAs seems a stretch. Clearly the authors have found evidence that some are, which is nice. But the way this is presented should be more appropriately written and justified.

Response: Lip formation completed at E11.5 and palatal development initiated at E11.5-E12.5. As I mentioned above, E12.5 was the time that all these facial prominences were easily discernable and could be repeatedly isolated, which was essential for the subsequent verification. However, I agree that “converging point” may not be the best word to describe this developmental stage and “turning point from facial development to palatal development” is more appropriate. Similarly, “skeletogenesis” was the wrong word and “osteogenic differentiation” was the accurate expression to describe the differentiation program at this stage. The expression was amended/highlighted in the relevant text.

8. The authors early on (line 91) state that multiple doses of atRA were used to induce full

penetrance CL/P. However, in the data presented (Fig 1), only one of the shown embryos has CL/P. One has CPO, and one purportedly has midfacial hypoplasia although this is not apparent from the image nor is there quantitative data to support this. For statements on penetrance, it is necessary to catalogue and present the incidence of all features in all embryos. It is not even mentioned how many embryos have been examined in this study. What percent have CL/P, CPO, CL, midfacial hypoplasia, mandibular clefts, neural tube defects, eye defects, etc? How do these compare to prior studies? (this goes to the robustness of the treatment regimen). The statement 'various incidences in all embryos' is incorrect as 'incidence refers to occurrence in the entire group not in specific embryos. It is better to say 'variable phenotypes in all embryos' but again this needs to be documented.

Response: The strategy of RA exposure used in this study was documented by Wang et al., in 2019. Full penetrance of CP was reported by them as well as was observed in our study. Palatal clefting can only be confirmed as early as E14.5-E15.5. The embryos presented in Fig 1C-E all had CP, which was not displayed in the images in 1C and 1E (displaying frontal view instead). The results presented in figure 1 were representative images of some of the malformations resulting from defective RA signaling in previous studies and were recapitulated in this research. Except for full penetrated CP, these malformations occur at various incidences. Given that the focus of this research is the altered gene expressions in craniofacial prominences induced by RA exposure, the detailed morphogenic characterizations have not been performed. Yes, I agree with the reviewer that the description of some of the results in figure 1 was misleading. This part was rephrased and highlighted in the relevant text.

9. mRNA-seq and miRNA-seq 'databases' - 'databases' is the wrong word here. Datasets is more accurate.

Response: Corrected and highlighted.

10. For the analysis of the RNA sequencing datasets, stating that 23173 (66.7%) were expressed in at least 4 samples is potentially misleading. Such statements are only useful if it relates to the 4 samples in that group (eg. all 4 WT sample or in all 4 treated embryos). Same applies to the miRNA seq statements. What % are in all 4 of each group? This is important to gauge reproducibility.

Response: Yes, the misleading description of “the genes detected in at least 4 samples” was deleted in the revised manuscript.

11. One line 168, the authors refer to Rxr but not which isoform(s) in the MdP despite stating isoform information for the MxP.

Response: Amended and highlighted in the revised manuscript.

12. Be more specific about the functions of the genes involved in Wnt signaling that are

upregulated. Are the positive effectors, downstream targets or inhibitors? (line 188). A latter statement (lines 225-227) makes it unclear, thus necessitating clarity earlier.
Response: Indicated the specific Wnt genes identified by others in their previous study.(line 240)

13. Again, the use of cleft lip as a keyword may not be the most appropriate given the analyses were done two days after fusion of the lip region when some of the known CL/P genes may be turned off.

Response: Yes, I agree that the focus of this study is the altered gene expression in facial prominence due to early RA exposure, not particularly related to cleft lip. Amended/highlighted in all relevant text.

14. Fig 1 needs to be accompanied by a summary table presenting the phenotypic variation and incidence of each presentation (eg. CL/P). Panel F is meant to be wildtype yet shows facial anomalies (hypoplasia), eye malformations, vascular anomalies and a neural tube defects!?! Whereas the treated embryo labeled as F' appears normal. An 'n' value for the number of embryos examined at each age needs to be mentioned in the legend and to address significance of all conclusions. The legend also does not mention any phenotype at all in B'. In E' the phenotype appears restricted to the mandible and therefore not CLP. The scale bar for the left side images (eg. A) is stated as 1mm. However, the length of the bar is significantly larger in A' suggesting the treated embryo is notably smaller than wildtype. Is this true? Or is the scalebar wrong? Is this an outlier or is it statistically significant? Is this size difference consistent at all stages of development? Panel H should be in the bottom right corner not above F and G.

Response: Again, as I mentioned above, the focus of this research is the altered gene expressions in craniofacial prominences induced by RA exposure. The detailed morphogenic characterizations and the incidences of each malformation have not been performed.

The scale bar in A' was correctly annotated. Variations in the sizes of embryos were observed in RA-treated embryos as well as in wild-type embryos. To the best of our knowledge, reduced overall sizes were not reported in RA-induced mouse embryos. We only considered the developmental stages of the embryos. Any differences in the genetic regulations resulted in the size differences would also be reflected in the transcriptome profiles, which might be further studied in the future.

Image of panel B' displayed a reduced growth of medial nasal process. Specified in the amended figure legend.

Panel F was a wild-type embryo and F' was a disfiguration, as annotated on top of the images.

Panel H was placed on top right because panel C-G were all displaying phenotypes at E15.5, as annotated on the left side of the images. We re-numbered the image panels to fix this problem.

15. In Fig 8 legend, it notes the abbreviation "PS" as palatoquadrate, which it is not - should be palatal shelf.

Response: The abbreviation should be “pq”. Amended and highlighted in the revised manuscript.

Reviewer #2 (Comments to the Authors (Required)):

In this article, authors investigated the spatiotemporal effect of excess RA exposure on regulating early development of facial prominences. The data showed that early exposure of RA inhibited the Wnt activities in E12.5/E13.5 mouse palatal shelves, but induced Wnt signaling and Wnt-related gene expression in mouse embryonic Frontonasal/Maxillary processes. A conserved regulatory network of miR-484-Fzd5 was identified, which may play critical roles in RA regulated craniofacial development. Foxn3 and Itgb1 can be induced by early RA-exposure. Besides, expression of BMP7 can also be affected by RA-exposure. This study is very interesting and may expand our knowledge about the role of RA in the development.

There are several comments for this article.

1. There are some typos need to be corrected.

Response: The English writing has been thoroughly checked by the authors as well as using Microsoft Word Editor. All corrections were highlighted in the manuscript.

2. Author found the regulatory network miR-484-Fzd5. What's the relationship between miR484-Fzd5 and Foxn3 or Itgb1 in excessive RA-exposure? More in vitro data should be provided to confirm the results of bioinformatic analysis.

Response: The RA-induced change in gene expression (miR-484, Fzd5, Foxn3, Itgb1, etc) has all been verified using qPCR at discrete facial prominences at various developmental stages, followed by functional assay both in vivo and in vitro.

3. What's the relationship between miR484-Fzd5 and BMP7, or Foxn3 / Itgb1 and BMP7 in excessive RA-exposure? More in vitro data should be provided to demonstrate the possible intrinsic connection.

Response: Fzd5 was confirmed as a new target of miR-484 in this study (refer to results displayed in Fig 4). Fzd5 is a receptor of canonical Wnt. Foxn3 and Itgb1 are transcriptionally regulated by TCF/LEF, which are transcription factors that respond to Wnt signaling. Foxn3 itself is also a transcription factor that regulate expression of Bmps, as shown by Samaan et al. The regulatory relationship between these factors was explained in the relevant text (line 257-259; line 301-303).

Further comments are welcome and please do not hesitate to contact us.

Best regards.

Sincerely,

Dr. Yi Zou

July 16, 2023

RE: Life Science Alliance Manuscript #LSA-2023-02013-TR

Dr. Yi Zou
Jinan University
Jinan University
Jinan University, No.601, West Huangpu Avenue , Guangzhou, Guangdong, China.
Guangzhou 510632
China

Dear Dr. Zou,

Thank you for submitting your revised manuscript entitled "RA-induced prominence-specific response resulted in distinctive regulation of Wnt and osteogenesis". We would be happy to publish your paper in Life Science Alliance pending final revisions necessary to meet our formatting guidelines.

- please address the remaining Reviewer 1's concerns
- please add the Twitter handle of your host institute/organization as well as your own or/and one of the authors in our system
- please add callouts for Figures 1H; S1A-K to your main manuscript text
- Data Availability section is present but incomplete: please add your mRNA and miRNA sequencing data accession codes

A. FINAL FILES:

B. MANUSCRIPT ORGANIZATION AND FORMATTING:

Sincerely,

Reviewer #1 (Comments to the Authors (Required)):

The authors have, for the most part, responded well. However some issues still persist. For example, although the authors have correct most of the English issues, there are still many sections where the tense is wrong or an incorrect (or not the best) word choice has been used. While the instances are too numerous to list individually, some examples include: line 51 (overlying "epidermis" - it is overlying "ectoderm" at these early stages of development. A proper epidermis, which arises from the ectoderm, does not appear until later in development); line 58 (instead of "lateral outgrowths", "mediolateral outgrowths" would be more accurate); line 37 ("progenitors" should just be "progenitor"); line 39-40 is in past tense but should not be. Line 113 - it would be more appropriate to use ug/g instead of mg/kg (even though meaning the same) since mice are measured in grams not kilograms. Line 126 - the description as "casualness" is poor. Simply just saying technical variance in sampling would suffice.

The sentence on line 159-162 seems to be based on other public data not the authors own data in the manuscript. This should be made more clear and cite the prior study as needed.

On page 9, the authors mention that heterogeneity of the samples with respect to tissue composition but fail to mention the significant heterogeneity due to sex. The authors respond to this issue raised in the first review round and acknowledge it but claim it does not matter. I beg to differ. The authors MUST indicate that their data comes from control and treated groups that are made up of different proportions of each sex (and provide those compositions). I still believe this will affect the numbers of differentially expressed genes. If the authors insist on ignoring this as a major confounder, then they need to at least be transparent and state that the sex composition of embryos that were pooled and that they recognize this could have an impact on the interpretation of their results.

The sentence on line 169-170 is very poor English. I would suggest perhaps rewording to something like: "with statistical significance were functionally annotated and enrichment analysis undertaken with no fold change cutoffs."

Line 215/216 - the description of "the cis-regulatory element of RARE" is redundant - RARE stands for (cis)-retinoic acid response element.

Line 219 - "that were induced" would be better as "and then" induced to indicate the treatment was AFTER the transfection.

Line 242 - "tempospatial" should be "temporospatial"

Line 305 - "thereby" is not the correct word to use here. "therefore" would be more appropriate - very different emphasis.

Finally, the first paragraphs of the Discussion (and to a lesser degree the second paragraph) needs to be completely rewritten - it reads like there is little flow to thought, just a mix of points. It could be a lot more concise and focused.

Line 628 - "capcity" spelling is incorrect.

July 20, 2023

RE: Life Science Alliance Manuscript #LSA-2023-02013-TR

Dear editor,

Thanks for the helpful comments from both the reviewer and the editorial office. Corrections have been made according to the suggestions. Any further questions are welcome and please do not hesitate to contact us. Our responses are presented below:

Dear Dr. Zou,

Thank you for submitting your revised manuscript entitled "RA-induced prominence-specific response resulted in distinctive regulation of Wnt and osteogenesis". We would be happy to publish your paper in Life Science Alliance pending final revisions necessary to meet our formatting guidelines.

-please address the remaining Reviewer 1's concerns

The remaining concerns of reviewer 1 were addressed below

-please add the Twitter handle of your host institute/organization as well as your own or/and one of the authors in our system

Added.

-please add callouts for Figures 1H; S1A-K to your main manuscript text

The callouts for Fig 1H and Fig S1A-K were presented in the main manuscript text (line 827 to 829; line 924 to 928)

-Data Availability section is present but incomplete: please add your mRNA and miRNA sequencing data accession codes

Accession codes were included.

Reviewer #1 (Comments to the Authors (Required)):

The authors have, for the most part, responded well. However some issues still persist. For example, although the authors have correct lost of the English issues, there are still many sections where the tense is wrong or an incorrect (or not the best) word choice has been used. While the instances are too numerous to list individually, some examples include: line 51 (overlying "epidermis" - it is overlying "ectoderm" at these early stages of development. A proper epidermis, which arises from the ectoderm, does not appear until later in development); line 58 (instead of "lateral outgrowths", "mediolateral outgrowths" would be more accurate); line 37 ("progenitors" should just be "progenitor"); line 39-40 is in past tense but should not be. Line 113 - it would be more appropriate to use ug/g instead of mg/kg (even though meaning the same) since mice are measured in grams not kilograms. Line 126 - the description as "casualness" is poor. Simply just saying technical variance in sampling would suffice.

All the above-mentioned spelling and grammar errors, along with others not mentioned, have been corrected and highlighted.

The sentence on line 159-162 seems to be based on other public data not the authors own data in the manuscript. This should be made more clear and cite the prior study as needed.

Ref was included.

On page 9, the authors mention that heterogeneity of the samples with respect to tissue composition but fail to mention the significant heterogeneity due to sex. The authors respond to this issue raised in the first review round and acknowledge it but claim it does not matter. I beg to differ. The authors MUST indicate that their data comes from control and treated groups that are made up of different proportions of each sex (and provide those compositions). I still believe this will affect the numbers of differentially expressed genes. If the authors insist on ignoring this as a major confounder, then they need to at least be transparent and state that the sex composition of embryos that were pooled and that they recognize this could have an impact on the interpretation of their results.

The heterogeneity in the sample sexes were specified (line 166-168).

The sentence on line 169-170 is very poor English. I would suggest perhaps rewording to something like: "with statistical significance were functionally annotated and enrichment analysis undertaken with no fold change cutoffs."

Amended and highlighted.

Line 215/216 - the description of "the cis-regulatory element of RARE" is redundant - RARE stands for (cis)-retinoic acid response element.

The redundant term was deleted.

Line 219 - "that were induced" would be better as "and then" induced to indicate the treatment was AFTER the transfection.

Amended and highlighted.

Line 242 - "tempospatial" should be "temporospatial"

Amended and highlighted.

Line 305 - "thereby" is not the correct word to use here. "therefore" would be more appropriate - very different emphasis.

Amended and highlighted.

Finally, the first paragraphs of the Discussion (and to a lesser degree the second paragraph) needs to be completely rewritten - it reads like there is little flow to thought, just a mix of points. It could be a lot more concise and focused.

Line 344-351 was repetitive and deleted, given that most of the facts have been addressed in the introduction. It also helped to improve the organization of the first paragraph. The rest of the first paragraph was rewritten to improve the flow of content.

Line 628 - "capcity" spelling is incorrect.

Amended and highlighted.

Warm regards.

Dr. Yi Zou

July 27, 2023

RE: Life Science Alliance Manuscript #LSA-2023-02013-TRR

Dr. Yi Zou
Jinan University
Guangzhou 510632
China

Dear Dr. Zou,

Thank you for submitting your Research Article entitled "RA-induced prominence-specific response resulted in distinctive regulation of Wnt and osteogenesis". It is a pleasure to let you know that your manuscript is now accepted for publication in Life Science Alliance. Congratulations on this interesting work.

DISTRIBUTION OF MATERIALS:

Again, congratulations on a very nice paper. I hope you found the review process to be constructive and are pleased with how the manuscript was handled editorially. We look forward to future exciting submissions from your lab.

Sincerely,
